# DIFFUSION-DFL: DECISION-FOCUSED DIFFUSION MODELS FOR STOCHASTIC OPTIMIZATION

**Zihao Zhao**
School of Computational Science and Engineering
Georgia Institute of Technology
zzhao628@gatech.edu

**Christopher Yeh**
Department of Computing and Mathematical Sciences
California Institute of Technology
cyeh@caltech.edu

**Lingkai Kong**
School of Engineering and Applied Sciences
Harvard University
lingkaikong@g.harvard.edu

**Kai Wang**
School of Computational Science and Engineering
Georgia Institute of Technology
kwang692@gatech.edu

## ABSTRACT

Decision-focused learning (DFL) integrates predictive modeling and optimization by training predictors to optimize the downstream decision target rather than merely minimizing prediction error. To date, existing DFL methods typically rely on deterministic point predictions, which are often insufficient to capture the intrinsic stochasticity of real-world environments. To address this challenge, we propose the first diffusion-based DFL approach, which trains a diffusion model to represent the distribution of uncertain parameters and optimizes the decision by solving a stochastic optimization with samples drawn from the diffusion model. Our contributions are twofold. First, we formulate diffusion DFL using the reparameterization trick, enabling end-to-end training through diffusion. While effective, it is memory and compute-intensive due to the need to differentiate through the diffusion sampling process. Second, we propose a lightweight score function estimator that uses only several forward diffusion passes and avoids backpropagation through the sampling. This follows from our results that backpropagating through stochastic optimization can be approximated by a weighted score function formulation. We empirically show that our diffusion DFL approach consistently outperforms strong baselines in decision quality. The source code for all experiments is available at https://github.com/GT-KOALA/Diffusion_DFL.

## 1 INTRODUCTION

Many real-life decision-making tasks require selecting actions that minimize a cost function involving unknown, context-dependent parameters. These parameters must often be predicted from observed features. For example, in supply chain management, future product demand must be estimated before deciding how much inventory to order (Tang & Nurmaya Musa, 2011). A common approach is the predict-then-optimize pipeline, where a predictive model is first trained using a loss function such as mean squared error (MSE), and the resulting predictions are then passed to an optimization solver to guide decisions. While simple and widely adopted, this two-stage method can be misaligned with the true objective: minimizing decision cost. In particular, lower prediction error does not always lead to higher-quality decisions (Bertsimas & Kallus, 2020; Elmachtoub & Grigas, 2022).

Decision-focused learning (DFL) addresses this misalignment by integrating the prediction and optimization stages into a single end-to-end framework (Donti et al., 2017; Wilder et al., 2019; Mandi et al., 2024). Unlike the two-stage approach, DFL trains the prediction model specifically to improve decision outcomes, often resulting in solutions with lower regret. However, most existing DFL methods rely on point (deterministic) predictions as inputs to the optimization layer, despite the fact that in many real-world scenarios, the underlying parameters are inherently uncertain and may follow complex distributions. Ignoring this uncertainty can lead to overconfident models and degraded decision quality (Kochenderfer et al., 2015).

In this work, we introduce a novel DFL approach that leverages diffusion probabilistic models to capture the environment uncertainty in an end-to-end fashion. Here, we use a conditional diffusion model (Tashiro et al., 2021) to represent the distribution of uncertain parameters given contextual features. The advantage of integrating a diffusion model into DFL is that, unlike simple distribution predictions (e.g., Gaussian), diffusion models can capture multi-modal or complex distributions. However, the sequential sampling procedure of diffusion models introduces a challenge when training a diffusion model end-to-end for stochastic optimization. To address this, we develop two algorithms: reparameterization and score function. First, the reparameterization trick is a common approach that expresses a random sample as a deterministic function of the model parameters and some noise, and we can backpropagate through sampled prediction to solve the DFL problem.

However, this approach can be very costly in memory and computation because it requires differentiating (and therefore tracking gradients) through the diffusion sampling process. To address this, we introduce a lightweight score function estimator that avoids differentiating through the sampling process. Specifically, we use a score function surrogate to approximate the gradient of the diffusion predictor and plug it into the KKT (Karush-Kuhn-Tucker) implicit-differentiation approach to obtain the total derivative of the decision objective. In addition, we further mitigate the high variance that arises from using only score functions for a few steps by employing a tailored importance sampling strategy. Specifically, instead of sampling diffusion timesteps uniformly, we assign sampling probabilities proportional to the gradient magnitude at each timestep, which keeps the estimator unbiased while substantially reducing its variance.

We evaluate our proposed methods in various applications, including (synthetic) product allocation, energy scheduling, and stock portfolio optimization. Experimental results show that our diffusion DFL methods consistently outperform all baselines, with more gains on larger-scale problems. Moreover, the score function estimator achieves decision quality comparable to that of the reparameterization method, while significantly reducing GPU memory usage from 60.75 GB to 0.13 GB.

The contributions of this paper are the following:

- We introduce the first DFL method that uses diffusion models to capture the downstream uncertainty and employs the reparameterization trick for end-to-end gradient estimation.
- We propose a lightweight score function estimator that avoids backpropagating the reversing process in the reparameterization method, significantly reducing memory and computation cost.
- We evaluate our methods in three real-world optimization tasks and observe consistent improvements over strong baselines.

## 2 RELATED WORKS

**Decision-focused learning.** DFL is a growing and increasingly influential approach that trains models end-to-end to directly optimize decision quality rather than minimize prediction error (Donti et al., 2017; Wilder et al., 2019; Mandi et al., 2024). Despite the success in aligning learning objectives with decision-making, a limitation of most existing DFL methods is that they typically rely on **deterministic point predictions** of uncertain parameters (Wilder et al., 2019; Shah et al., 2022). By ignoring distributional uncertainty, deterministic point predictions cannot represent the full outcomes and may lead to lower decision quality (Wang et al., 2025). Empirically, classic DFL was observed to struggle in high-dimensional and risk-sensitive real-world settings with significant uncertainty (Mandi et al., 2022).

Therefore, the gap in uncertainty modeling motivates the need for more comprehensive DFL with *stochastic predictions*, where several works have started integrating uncertainty awareness into the DFL pipeline (Silvestri et al., 2026; Wang et al., 2025; Shariatmadar et al., 2025; Jeon et al., 2025). For instance, Wang et al. (2025) proposes a generative DFL approach (Gen-DFL) based on normalizing flow models as the predictor. However, normalizing flows require a bijective network architecture, which restricts the expressiveness of the stochastic predictor.

In this paper, we propose using diffusion models (Ho et al., 2020) as a more expressive predictor. By leveraging diffusion models in the DFL paradigm, our approach extends DFL by predicting an accurate full distribution of the unknown parameters, which addresses the overconfidence of deterministic optimization and better aligns with downstream decision-making needs.

Orthogonally, Silvestri et al. (2026) propose using score function gradient estimation to extend DFL to settings with non-differentiable optimization solvers. However, their approach relies on predictors for which the score admits a closed-form expression. In contrast, the score cannot be computed in closed form for diffusion models. To address this, we approximate the score using the gradient of the EBLO as a surrogate and provide a theoretical bound on the approximation error.

**Diffusion models in optimization.** Diffusion probabilistic models have achieved great success in modeling high-dimensional data distributions in recent years (Sohl-Dickstein et al., 2015; Song & Ermon, 2019; Dhariwal & Nichol, 2021). Originally popularized for image generation and related structured outputs, its ability to capture multi-modal and high-variety distributions has made it attractive beyond vision tasks, such as combinatorial optimization (Sun & Yang, 2023; Sanokowski et al., 2025) and black-box optimization (Krishnamoorthy et al., 2023; Kong et al., 2025).

To our best knowledge, however, no prior work has integrated diffusion models into a predict-then-optimize learning pipeline for decision tasks. This paper is the first to harness diffusion models in an end-to-end DFL framework. By using a conditional diffusion model, we can learn a rich distribution over the uncertain inputs and then propagate this uncertainty through to the downstream decision via gradient-based training (score function and reparameterization). This approach combines the strengths of expressive generative modeling and DFL to improve decision quality under uncertainty.

## 3 PROBLEM STATEMENT

### 3.1 DECISION-FOCUSED LEARNING

We consider a general predict-then-optimize setting (Donti et al., 2017; Elmachtoub & Grigas, 2022), where the goal is to make decisions under uncertainty about a key problem parameter. Given a feature vector $x \in \mathcal{X}$ and a prediction of an unknown parameter $y^* \in \mathcal{Y}$, the decision-maker selects $z \in \mathbb{R}^d$ to minimize a decision loss function $f : \mathcal{Y} \times \mathbb{R}^d \to \mathbb{R}$, which measures the cost of applying decision $z$ when the true parameter is $y^*$. We assume a joint distribution $\mathcal{D}$ over $(x, y^*)$ pairs.

DFL integrates prediction and optimization into a unified framework. The goal is to learn a decision function $z_\theta^* : \mathcal{X} \to \mathbb{R}^d$, parameterized by $\theta$, that minimizes the expected decision loss,

$$\min_\theta F(\theta) := \mathbb{E}_{(x,y^*)\sim\mathcal{D}}[f(y^*, z_\theta^*(x))].$$

The decision $z_\theta^*(x)$ is typically obtained by solving an optimization problem involving a prediction of the uncertainty parameter. Most DFL methods (Mandi et al., 2024) use a deterministic point prediction $y_\theta(x)$ of the uncertain parameter $y^*$:

$$z_\theta^*(x) = \arg\min_z f(y_\theta(x), z), \quad \text{s.t. } Gz \le h, \ Az = b,$$

where $G \in \mathbb{R}^{n \times d}, h \in \mathbb{R}^n, A \in \mathbb{R}^{p \times d}, b \in \mathbb{R}^p$ are constraint problem coefficients[1].

In contrast, we consider a probabilistic model $P_\theta(\cdot \mid x)$ for the uncertain parameter $y^*$ and let $z_\theta^*(x)$ be the solution to a stochastic optimization problem:

$$z_\theta^*(x) = \arg\min_z \mathbb{E}_{y\sim P_\theta(\cdot|x)}[f(y, z)], \quad \text{s.t. } Gz \le h, \ Az = b. \tag{1}$$

We aim to learn the model parameter $\theta$ such that $z_\theta^*$ minimizes the expected decision loss $F(\theta)$. By the chain rule, the derivative of $F$ is

$$\frac{dF(\theta)}{d\theta} = \mathbb{E}_{(x,y^*)\sim\mathcal{D}}\left[\frac{\partial f(y^*, z_\theta^*(x))}{\partial z} \frac{dz_\theta^*(x)}{d\theta}\right].$$

However, computing this gradient (specifically, the $\frac{dz_\theta^*}{d\theta}$ term) is challenging because $z_\theta^*$ is implicitly defined by a nested optimization problem. A common solution is to differentiate the KKT system that implicitly defines $z_\theta^*$ w.r.t. $\theta$ (Amos & Kolter, 2017). Another crucial point is the selection of the stochastic predictor in DFL, which in the paper we choose to use diffusion models to represent $P_\theta$.

---

[1] We consider affine constraints in our main paper for simplicity. The extension from affine constraints to general convex constraints follows a similar derivation as in the linear case.

## 3.2 Diffusion Probabilistic Model

To generate complex multi-modal and high-dimensional distributions, diffusion probabilistic models (Ho et al., 2020) are a promising approach. It couples a fixed noising chain with a learned reverse denoising chain. Let $y_0 \in \mathbb{R}^d$ denote a sample from the real data distribution $q(y_0)$ and $\{\beta_t \in (0,1)\}_{t=1}^T$ denote the noise schedule. Define $\alpha_t = 1 - \beta_t$ and $\bar{\alpha}_t = \prod_{i=1}^t \alpha_i$. The *forward process q* adds Gaussian noise at each step $t$ to $y_1$ through $y_T$:

$$q(y_t \mid y_{t-1}) = \mathcal{N}(y_t; \sqrt{1-\beta_t}y_{t-1}, \beta_t I), \quad t = 1, \dots, T,$$

which guarantees that $q(y_T \mid y_0)$ becomes nearly standard normal distribution as $T \to \infty$ with common schedules ($\bar{\alpha}_T \to 0$). Note that $y_t$ can be equivalently sampled without iterating through intermediate time steps: $y_t = \sqrt{\bar{\alpha}_t}y_0 + \sqrt{1-\bar{\alpha}_t}\epsilon$, where $\epsilon \sim N(0,I)$ is a Gaussian noise.

In the *reverse process p*, the diffusion model predicts the unknown added noise by

$$p_\theta(y_{t-1} \mid y_t) = \mathcal{N}(y_{t-1}; \mu_\theta(y_t, t), \sigma_t^2 I), \tag{2}$$

whose mean $\mu_\theta(\cdot, t)$ is parameterized by a neural network predictor and variance is either fixed ($\sigma_t^2 = \beta_t$) or learned. The combination of $p$ and $q$ is equivalent to a hierarchical variational auto-encoder (Vahdat & Kautz, 2020), and thus can be optimized by using the evidence lower bound (ELBO) as the loss function (Hoffman & Johnson, 2016).

**Conditional Diffusion Model.** Throughout this paper, $x$ denotes contextual features, and every transition probability is conditioned on $x$ (Tashiro et al., 2021). We use $P_\theta(\cdot|x)$ for the diffusion model's conditional distribution for generated data and $p_\theta(y_{t-1} \mid y_t, x)$ for its Markov transitions.

## 4 Stochastic Optimization and Reparameterization estimator

Real-world decision problems often face significant uncertainty in their parameters. Optimizing with a stochastic predictor (e.g., diffusion model) yields better results than deterministic optimization, by explicitly modeling the uncertainty and optimizing the expected cost. Figure 1 illustrates a simple example: any deterministic solution ends up at an extreme decision with a higher expected cost, while the stochastic solution averages costs across likely outcomes and selects an interior decision with a lower expected cost.

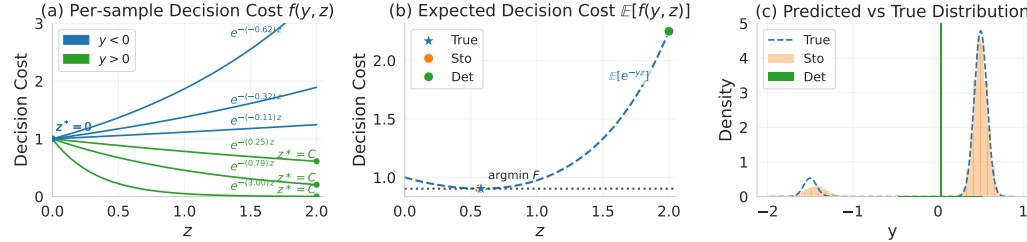

Figure 1: A comparison of deterministic vs. stochastic optimization with cost function $\exp(-yz)$, as described in Section 6.1. **(a)** Each curve represents a cost function given a sample $y$. For any fixed $y$, the deterministic optimization decision lies at one of the boundaries ($z^* = 0$ or $z^* = C$). **(b)** When averaging the cost function over many samples of $y$, the stochastic optimization decision lies in the interior of the feasible region instead of on the boundary. Thus, any deterministic optimization decision is suboptimal. **(c)** A probabilistic (diffusion) model captures a distribution over $Y$ that closely resembles the true bimodal distribution.

**Solving stochastic DFL.** Formally, in the stochastic case, the optimality condition for the decision problem must consider an expectation. The stationarity condition for decision problem Eq. 1 becomes

$$\nabla_z \mathcal{L}(\theta, z^*, \lambda^*, \nu^*; x) = \mathbb{E}_{y \sim P_\theta(\cdot|x)}[\nabla_z f(y, z)] + G^\top \lambda^* + A^\top \nu^* = 0,$$

where $\mathcal{L}$ denotes the Lagrangian. Note that the dependency on $\theta$ in the stationarity condition is in the distribution. Therefore, we need to handle this dependency carefully while differentiating the KKT

system with respect to $\theta$:

$$\underbrace{\frac{\partial}{\partial \theta}(\nabla_z \mathcal{L}(\theta, z^*, \lambda^*, \nu^*; x))}_{\text{distributional gradient}} = \frac{\partial}{\partial \theta}(\mathbb{E}_{y \sim P_\theta(\cdot|x)}[\nabla_z f(y, z)] + G^\top \lambda^* + A^\top \nu^*) = 0. \tag{3}$$

To resolve the dependence of both the predictive distribution $P_\theta(y \mid x)$ and the decision $z^*$ on $\theta$, we first adopt the *reparameterization trick* (Kingma & Welling, 2014) for the diffusion model. From Section 3.2, recall that the diffusion sampling process introduces Gaussian noise at each step. Thus, we can reparameterize the reverse process by fixing all the random draws (Gaussian noises). Formally, a sample $y \sim P_\theta(y \mid x)$ can be expressed as a transformation $y = R(\epsilon, \theta \mid x)$ of a base Gaussian noise sample $\epsilon \sim P(\epsilon)$, where $R$ is differentiable in $\theta$. This makes the diffusion sampling a deterministic function of $\theta$. Then we have

$$\nabla_\theta \mathbb{E}_{y \sim P_\theta(\cdot|x)}[f(y, z)] = \mathbb{E}_{\epsilon \sim P(\epsilon)}[(\nabla_\theta R(\epsilon, \theta|x))^\top \nabla_y f(y, z)]. \tag{4}$$

Next, we incorporate this into the optimization. Following Eq. 3, we can formalize a KKT system that contains derivatives of $z_\theta^*, \lambda^*, \nu^*$ and $\nabla_\theta \mathbb{E}_{y \sim P_\theta(\cdot|x)}[f(y, z)]$. Plugging the reparameterized gradient estimator into the KKT system, we can solve for $\frac{dz_\theta^*}{d\theta}$ and then obtain the total derivative of the final objective $F$ by multiplying $\frac{dF}{dz_\theta^*}$ (Donti et al., 2017) (proof can be found in Appendix A.1):

$$\frac{dF}{d\theta} = -\begin{bmatrix} \frac{dF}{dz_\theta^*} \\ 0 \\ 0 \end{bmatrix}^\top \begin{bmatrix} H & G^\top & A^\top \\ D(\lambda^*)G & D(Gz_\theta^* - h) & 0 \\ A & 0 & 0 \end{bmatrix}^{-1} \begin{bmatrix} \mathbb{E}_{\epsilon \sim P(\epsilon)}[(\nabla_\theta R(\epsilon, \theta|x))^\top \nabla_{zy}^2 f(y, z_\theta^*)] \\ 0 \\ 0 \end{bmatrix}, \tag{5}$$

where $H = \mathbb{E}_{y \sim P_\theta(\cdot|x)}[\nabla_{zz}^2 f(y, z_\theta^*)]$ is the Hessian of the Lagrangian with respect to $z$, and $D(v)$ denotes a diagonal matrix with $v$ on its diagonal. In practice, one can sample $\epsilon$ from a certain distribution (e.g., Gaussian) multiple times to estimate the expectation and then obtain the gradient. This gives us reparameterization-based diffusion DFL using Eq. 5 to run stochastic DFL optimization.

## 5 SCORE FUNCTION ESTIMATOR

A major obstacle to implementing the total gradient (Eq. 5) is the need to backpropagate through the diffusion sampling process. In most cases, the diffusion model's generative process is complex and multi-step (e.g., 1000 steps), which makes backpropagating through all those steps memory-intensive and prone to instability. To address this, we propose a **score function**[2] gradient estimator for the diffusion model, which circumvents explicit backpropagation through all sampling steps. The key idea is to rewrite the Jacobian $\nabla_\theta y$ in terms of the score $\nabla_\theta \log P_\theta(y \mid x)$, and then approximate the score with the diffusion model's ELBO training loss.

### 5.1 TRANSFORM THE JACOBIAN INTO SCORE FUNCTION

We begin by rewriting the gradient of expectation as an expectation of a score function using the *log-trick* (Mohamed et al., 2020). Formally, if $y \sim P_\theta(\cdot|x)$ and $f$ is a function not dependent on $\theta$, then by the log-derivative trick we have

$$\nabla_\theta \mathbb{E}_{y \sim P_\theta(\cdot|x)}[f(y, z)] = \mathbb{E}_{y \sim P_\theta(\cdot|x)}[f(y, z) \cdot \nabla_\theta \log P_\theta(y \mid x)]. \tag{6}$$

Intuitively, instead of differentiating the output $y$ through each diffusion step, we only need to compute the gradient for the final log-likelihood, which avoids the need to differentiate through the diffusion sampling process and yields an efficient estimator for the gradient.

Then, one remaining difficulty is that directly computing the exact $\nabla_\theta \log P_\theta(y \mid x)$ is complicated in practice because $P_\theta(y \mid x)$ is defined as the marginal probability of $y$ after integrating out the latent diffusion trajectory. To obtain a computationally efficient estimator, we use the diffusion model's

---

[2]In this paper, score function refers to the statistical score $\nabla_\theta \log P_\theta(y \mid x)$ (gradient of log-likelihood w.r.t. model parameters), as opposed to Stein's score $\nabla_y p(y_t \mid y_{t-1}, x)$ often used in the diffusion literature.

training objective as a surrogate for the log-likelihood. Specifically, diffusion models are typically trained by maximizing an ELBO that lower-bounds the log-likelihood:

$$\log P_\theta(y_0) = \log \int p_\theta(y_0|y_1)\, p_\theta(y_1|y_2)\cdots p_\theta(y_{T-1}|y_T)\, p_\theta(y_T)\, dy_{1:T}$$

$$= \log \mathbb{E}_{y_t \sim q(y_t|y_{t-1})\forall t\in[T]} \left[ \prod_{t=1}^{T} \frac{p_\theta(y_{t-1}|y_t)}{q(y_t|y_{t-1})} p_\theta(y_T) \right]$$

$$\geq \mathbb{E}_{y_t \sim q(y_t|y_{t-1})\forall t\in[T]} \left[ \sum_{t=1}^{T} \log \frac{p_\theta(y_{t-1}|y_t)}{q(y_t|y_{t-1})} + \log p_\theta(y_T) \right] =: \mathrm{ELBO}(y_0;\theta), \quad (7)$$

where the inequality is due to Jensen's. To approximate $\nabla_\theta \log P_\theta(y_0 \mid x)$ conditioned on $x$, we use the gradient of the conditional ELBO loss as a surrogate:

$$\nabla_\theta \log P_\theta(y_0 \mid x) \approx \nabla_\theta \mathrm{ELBO}(y_0 \mid x; \theta). \tag{8}$$

To motivate this approximation, we provide a theoretical justification for the surrogate ELBO gradient.

**Proposition 5.1** (ELBO gradient approximation error). *Let $s_\theta(z; y_0) := \nabla_\theta \log p_\theta(y_0, z)$ denote the joint score function. If $\sup_w \|s_\theta(z; y_0)\| \leq B(\theta, y_0)$ for some finite upper bound $B(\theta, y_0) > 0$, then*

$$\|\nabla_\theta \log P_\theta(y_0) - \nabla_\theta \mathrm{ELBO}(y_0;\theta)\| \leq \sqrt{2}B(\theta,y_0)\sqrt{\mathrm{KL}(q(y_{1:T} \mid y_0)\|p_\theta(y_{1:T} \mid y_0))}.$$

See Appendix A.4 for proof and more details. Consequently, whenever the variational posterior (diffusion reverse process) is a good approximation of the target distribution (diffusion forward process), so that the KL divergence term is small, the ELBO gradient is guaranteed to be close to the true score. This justifies our use of $\nabla_\theta \mathrm{ELBO}(y_0;\theta)$ as a surrogate for $\nabla_\theta \log P_\theta(y_0)$.

In practice, we first sample a final output $y$ from the diffusion model given contextual features $x$. We then sample a subset of $k$ timesteps $\{t_1, t_2, \ldots, t_k\}$ ($k \ll T$) and run forward noising process $q$ to generate the trajectory $\{y_{t_1}, y_{t_2}, \ldots, y_{t_k}\}$. As in DDPM (Ho et al., 2020), we adopt the simplified form of ELBO $\approx \mathbb{E}_{t\sim[T],y_0,\epsilon_t}[\|\|\epsilon_t - \epsilon_\theta(y_t, t)\|^2]$. We evaluate the ELBO on the sampled trajectories and compute its gradient w.r.t. $\theta$ as an estimation of the true score.

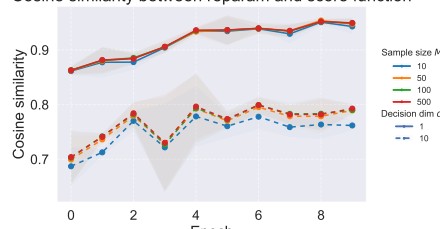

Figure 2: Cosine similarity between the reparameterization and score function gradient across different dimensions.

Empirical evidence suggests that the ELBO gradient closely tracks the true score, as shown in Figure 2, making Eq. 8 a reliable proxy in practice.

## 5.2 Overall Gradient for Score Function

By plugging the ELBO gradient approximation from Eq. 8 into Eq. 6, we can express the KKT conditions without using reparameterization and thus obtain the score function-based derivative:

$$\frac{dF}{d\theta} \approx - \begin{bmatrix} \frac{dF}{dz_\theta^*} \\ 0 \\ 0 \end{bmatrix}^\top \begin{bmatrix} H & G^\top & A^\top \\ D(\lambda^*)G & D(Gz_\theta^* - h) & 0 \\ A & 0 & 0 \end{bmatrix}^{-1} \begin{bmatrix} \mathbb{E}_{y\sim P_\theta(\cdot|x)}[\nabla_z f(y, z_\theta^*)(\nabla_\theta \mathrm{ELBO}(y|x;\theta))^\top] \\ 0 \\ 0 \end{bmatrix}.$$
$$(9)$$

**Practical algorithm – weighted ELBO gradient.** To compute the score surrogate in practice, we found it convenient to treat the total gradient as an importance-weighted form:

$$\frac{dF}{d\theta} \approx \frac{d}{d\theta} \mathbb{E}_{y\sim P_\theta(\cdot|x)} [ \underbrace{\mathrm{detach}[w_\theta(y)]}_{\text{importance weight, no grad in }\theta} \cdot \underbrace{\mathrm{ELBO}(y|x,\theta)}_{\text{1-step forward}} ], \tag{10}$$

where $w_\theta(y)$ is the importance weight simplified from Eq. 9 (see Appendix A.3 for complete form). This yields a *weighted-ELBO* gradient estimator: we treat $w_\theta(y)$ as a stop-gradient weight and only differentiate the ELBO w.r.t. $\theta$, greatly reducing computations. We implement the entire gradient computation as a user-friendly PyTorch autograd module: the forward pass returns the optimal decision $z^*$ (and $\lambda^*, \nu^*$), and the backward pass computes the gradient $\frac{dF}{d\theta}$ as derived above.

**Variance-reduction strategy.** While the score-function estimator is effective, a naive implementation of the weighted ELBO loss in Eq. 10 can suffer from high variance, leading to unstable training. In practice, we found that carefully designing the sampling strategy for the ELBO loss is crucial to obtaining low-variance and stable gradients. To reduce the variance, we utilize the method from Improved DDPM (Nichol & Dhariwal, 2021) for choosing diffusion steps. Specifically, instead of uniform sampling, we use *importance sampling* over timesteps with probability $p_t$ and weights $1/p_t$:

$$\nabla_\theta \text{ELBO}^{\text{IS}} = \mathbb{E}_{t \sim p_t} \left[ \frac{\nabla_\theta(\text{ELBO}_t)}{p_t} \right], \quad \text{where } p_t \propto \sqrt{\mathbb{E}[\|\nabla_\theta(\text{ELBO}_t)\|^2]} \text{ and } \sum_t p_t = 1.$$

This method remains unbiased, but the variance is minimized. In essence, this approach gives less weight to the early timesteps that have large gradients and more weight to later timesteps.

## 6 EXPERIMENTS

We evaluate the performance of our diffusion-based DFL approaches on a variety of tasks, comparing against several baseline methods. Specifically, we consider:

- **Two-stage predict-then-optimize baselines**: a deterministic MLP, a Gaussian probabilistic model, and a diffusion model trained to minimize prediction error (Elmachtoub & Grigas, 2022).
- **Deterministic DFL**: a deterministic MLP model with end-to-end DFL training (Donti et al., 2017).
- **Gaussian DFL** (both reparameterization and score function): a Gaussian probabilistic model with end-to-end stochastic DFL training; see details in Appendix A.6.
- **Offline Contextual Bandits**: solving the full-information contextual bandit using policy-based learning (Brandfonbrener et al., 2021). Note that this method ignores the constraints of the optimization. See details in Appendix A.7.
- **Diffusion DFL (ours)**: our diffusion model predictor, trained with either reparameterization or score-function gradient estimators.

Architectural and training details for all methods are summarized in Appendix A.8.

### 6.1 SYNTHETIC PRODUCT ALLOCATION TASK

In this example, we consider a factory that decides how much to manufacture for each of $d \in \mathbb{N}$ products. The parameter $Y \in \mathbb{R}^d$ represents the *profit margin* for each product, i.e., $Y_i$ is the profit per unit of product $i$; due to uncertainty in market conditions, $Y$ is uncertain. The factory's decision $z \in [0, C]^d$ represents how much of each product to manufacture, where $C$ is the maximum capacity for each product. For simplicity, we do not consider any contextual features $x$ in this example. That means DFL learns a distribution that generates $y$ that can minimize the decision objective.

Suppose that the factory has a risk-averse cost function $f(y, z) = \exp(-y^\top z)$, which indicates that the factory wants to put a larger weight on the product with higher profit[3] $Y_i$. Under uncertainty, the decision-maker seeks to minimize the *expected cost* by solving a stochastic optimization problem:

$$z_{\text{sto}}^* \in \arg\min_{z \in [0,C]^d} \mathbb{E}_{y \sim P_\theta(\cdot|x)}[\exp(-y^\top z)].$$

In this stochastic case, the optimal investment $z_{\text{sto}}^*$ typically lies in the interior of the feasible region, which balances the potential high reward of investing against the risk of losses.

**Experimental setup.** We simulate the uncertain parameter $Y$ drawn from a mixture of Gaussians,

$$Y_i \overset{\text{iid}}{\sim} p \cdot \mathcal{N}(a, \sigma^2) + (1 - p) \cdot \mathcal{N}(-b, \sigma^2).$$

Specifically, we set $p = 0.8, a = 1, b = 3, \sigma = 0.15, C = 2$. We train each model (deterministic, Gaussian, diffusion) on this distribution in a decision-focused manner (for DFL methods) or on pure regression (for two-stage), and evaluate the expected cost achieved by the resulting decision $z_\theta^*(x)$. We present the results of one product ($d = 1$) in Figure 1 and 10 products ($d = 10$) in Table 1.

---

[3]Here, we have ignored the degenerate case $y = 0$. To deal with the degenerate case, one could add a zero-centered bump function $c(y)$ to the objective $f(y, z)$.

## 6.2 POWER SCHEDULING TASK

In this experiment, we evaluate our method on a real-world energy scheduling problem from Donti et al. (2017). This task involves a 24-hour generation-scheduling problem in which the operator chooses $z \in \mathbb{R}^{24}$ (hourly generation). Given a realization $y$ of demand, the decision loss penalizes shortage and excess with asymmetric linear costs ($\gamma_s$ and $\gamma_e$) plus a quadratic tracking term; the decision must also satisfy a ramping bound $c_r$. Let $[v]_+ := \max(v, 0)$. We have the decision loss as the quadratic problem:

$$\min_z \ \mathbb{E}_{y \sim P_\theta(\cdot|x)}[f(y,z)] = \sum_{i=1}^{24} \mathbb{E}_{y \sim P_\theta(\cdot|x)}[\gamma_s[y_i - z_i]_+ + \gamma_e[z_i - y_i]_+ + \frac{1}{2}(z_i - y_i)^2],$$

$$\text{s.t. } |z_i - z_{i-1}| \le c_r \text{ for all } i \in \{1, 2, \dots, 24\}.$$

**Experimental setup.** We use more than 8 years of historical data from a regional power grid (PJM Interconnection, 2025). Feature $x$ includes the previous day's hourly load, temperature, next-day temperature forecasts, non-linear transforms (lags and rolling statistics), calendar indicators, and yearly sinusoidal features. Given $x$, the prediction model $P_\theta(\cdot|x)$ outputs a distribution over $y \in \mathbb{R}^{24}$. We report the test decision cost in Table 1 and a held-out horizon in Figure 7.

## 6.3 STOCK MARKET PORTFOLIO TASK

In this experiment, we apply our diffusion DFL approach to a financial portfolio optimization problem under uncertain stock returns. Here, the random vector $y \in \mathbb{R}^n$ represents the returns for the assets $n$ on the next day, and the decision $z \in \mathbb{R}^n$ represents the portfolio weights allocated to those assets. We consider a mean-variance trade-off decision loss: maximize expected return while keeping the risk (variance) low. This can be written as minimizing a loss that is a negative expected return plus a quadratic penalty on variance:

$$\min_z \ \mathbb{E}_{y \sim P_\theta(\cdot|x)}[f(y,z)] = \mathbb{E}_{y \sim P_\theta(\cdot|x)} \left[ \frac{\alpha}{2} z^\top y y^\top z - y^\top z \right], \quad \text{s.t.} \quad z^\top \mathbf{1} = 1, \ 0 \le z_i \le 1,$$

where $\alpha > 0$ is a risk parameter and constraints enforce that $z$ is a valid portfolio. In practice, the deterministic solution may concentrate heavily on a few assets and yield a low average return, whereas a stochastic approach can achieve higher returns by accounting for variance.

**Experimental setup.** We have daily prices and volumes spanning 2004-2017 and evaluate on the S&P 500 index constituents (Quandl WIKI dataset, 2025). The features $x \in \mathbb{R}^{28}$ include recent historical return, trading volume windows, and rolling averages. The immediate-return predictor $P_\theta(\cdot|x)$ is to predict the next day's price. We report the performance of different DFL baselines with 50 portfolios in Table 1 and other sizes of portfolios in Section 7.2.

Table 1: Results for different optimization tasks. Our two diffusion DFL methods achieve the best and second-best decision quality in all 3 tasks, significantly better than other baselines. **Bolded** values are the best in test task losses; underlined values are the 2nd-best. Mean $\pm$ standard error across 10 runs.

| Label / Method | Synthetic Example | | Power Schedule | | Stock Portfolio | |
| --- | --- | --- | --- | --- | --- | --- |
| | RMSE↓ | Task↓ | RMSE↓ | Task↓ | RMSE↓ | Task (%)↑ |
| *Two-stage (TS)* | | | | | | |
| Deterministic TS | $0.639_{\pm 0.00}$ | $1.987_{\pm 0.00}$ | $0.120_{\pm 0.00}$ | $41.239_{\pm 3.18}$ | $0.027_{\pm 0.00}$ | $0.04\%_{\pm 0.04}$ |
| Gaussian TS | $0.720_{\pm 0.00}$ | $1.272_{\pm 0.23}$ | $0.117_{\pm 0.00}$ | $5.580_{\pm 0.45}$ | $0.188_{\pm 0.03}$ | $0.10\%_{\pm 0.04}$ |
| Diffusion TS | $0.905_{\pm 0.00}$ | $0.393_{\pm 0.00}$ | $0.147_{\pm 0.00}$ | $7.901_{\pm 0.76}$ | $0.455_{\pm 0.00}$ | $0.13\%_{\pm 0.03}$ |
| *Offline Contextual Bandits* | | | | | | |
| Policy-based Learning | $0.873_{\pm 0.00}$ | $0.568_{\pm 0.02}$ | $4.712_{\pm 0.10}$ | $4.440_{\pm 0.17}$ | $0.064_{\pm 0.00}$ | $1.74\%_{\pm 0.41}$ |
| *Decision-focused learning (DFL)* | | | | | | |
| Deterministic | $0.640_{\pm 0.00}$ | $1.987_{\pm 0.00}$ | $4.997_{\pm 0.10}$ | $4.324_{\pm 0.25}$ | $0.032_{\pm 0.00}$ | $0.07\%_{\pm 0.00}$ |
| Gaussian Reparameterization | $0.707_{\pm 0.00}$ | $1.169_{\pm 0.03}$ | $4.525_{\pm 0.12}$ | $3.724_{\pm 0.05}$ | $0.189_{\pm 0.03}$ | $0.08\%_{\pm 0.03}$ |
| Gaussian Score Function | $0.708_{\pm 0.00}$ | $1.132_{\pm 0.00}$ | $4.713_{\pm 0.15}$ | $4.087_{\pm 0.06}$ | $0.187_{\pm 0.03}$ | $0.14\%_{\pm 0.05}$ |
| Diffusion Reparameterization | $0.852_{\pm 0.01}$ | $\underline{0.365}_{\pm 0.00}$ | $3.141_{\pm 0.06}$ | $\mathbf{3.152}_{\pm 0.03}$ | $0.063_{\pm 0.00}$ | $\mathbf{4.17\%}_{\pm 0.24}$ |
| Diffusion Score Function | $0.849_{\pm 0.09}$ | $\mathbf{0.362}_{\pm 0.00}$ | $2.893_{\pm 0.03}$ | $\underline{3.171}_{\pm 0.02}$ | $0.067_{\pm 0.00}$ | $\underline{3.98\%}_{\pm 0.31}$ |

## 7 Discussion of Experimental Results and Ablation Study

### 7.1 Discussion of Results in Table 1

**Two-stage vs. DFL.** As shown in Table 1, across all three experiment tasks, we find that end-to-end DFL leads to better downstream decisions than the conventional two-stage approach. Conventional two-stage methods minimize RMSE during training, but this often leads to poor downstream decisions. In contrast, all variants of DFL directly minimize the decision cost during training and thus achieve lower decision costs.

**Offline Contextual Bandits vs. DFL.** Table 1 shows that the policy-based offline bandit method can outperform two-stage approaches on tasks such as power scheduling and portfolio tasks, since it directly optimizes the downstream task objective rather than training a predictor with a surrogate loss. It is also competitive with deterministic and Gaussian DFL variants. One reason is that, in some settings (such as the synthetic task), DFL with deterministic optimization (the Gaussian case can be written in closed-form as a deterministic optimization) fails to represent multi-modal distributions on uncertain parameters and is therefore a fundamentally poor policy class that does not include the optimal policy. An expressive enough OCB policy class, on the other hand, can learn a strong context-to-action policy.

Nevertheless, Diffusion-DFL remains consistently superior to offline contextual bandits: by incorporating the known optimization structure and coupling a flexible generative model with the downstream solver, it can better capture complex decision distributions and deliver higher-quality decisions.

**Deterministic vs. Stochastic Optimization.** Our results show that stochastic DFL methods outperform deterministic DFL in terms of decision quality on every task. By modeling uncertainty, stochastic predictors enable the decision optimization to account for risk and variability in outcomes. For instance, in the portfolio experiment, the deterministic DFL yields only $0.07\%$ return, whereas a Gaussian DFL modestly improves that, and our diffusion DFL achieves nearly $4\%$ average return. These gains come from the stochastic models' ability to predict uncertainty: instead of committing to a point prediction of $y$, the stochastic DFL produces decisions for a range of possible outcomes.

**Benefits of Diffusion DFL.** Among the stochastic approaches, including baselines using Gaussian models, our diffusion DFL method consistently delivers the best decision performance. In particular, the diffusion model's strength is the capacity to capture complex, multi-modal outcome distributions that a simple parametric Gaussian cannot represent. The Gaussian DFL sometimes falls short of the optimal decision quality. The diffusion model, on the other hand, can represent more intricate distributions of $y$, leading to decisions that better reflect complex scenarios.

### 7.2 Ablation Study

**Comparison Cost for Reparameterization and Score function.** A key finding from our ablation study is the computational advantage of score-function approach over the reparameterization. Here, we measure the trade-off between training cost and the final decision performance for different gradient estimators and sampling budgets.

In **??** (a), we see that all variants reach similar final performance on the test set, indicating that even using as few as 50 samples is sufficient to optimize the decision quality accurately. Figure 4 plots the GPU memory cost alongside the final test loss. The reparameterization method is very computationally expensive, requiring about 60 GB of GPU memory for backpropagating through all diffusion steps. In contrast, the score-function with 50 samples achieves virtually the same test loss as the reparameterization method (difference within 0.02) while using an order of magnitude less memory. Even with 10 samples, though slightly worse in loss, it still outperforms the deterministic baseline and uses a tiny fraction of the compute. These results validate that the score-function approach retains the decision-quality benefits of diffusion DFL while dramatically cutting computational requirements, making diffusion DFL practical even for complex problems.

**Gradient variance reduction.** As discussed in Section 5.2, using the score function estimator allows us to avoid backpropagating through the entire diffusion sampling process by only sampling a limited number of diffusion timesteps per update. The reason behind this is that a naive implementation, sampling timesteps uniformly at random, would yield a very high variance in the gradient estimates,

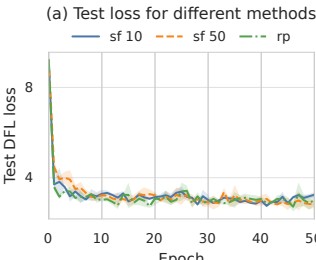 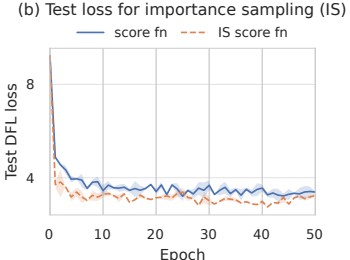 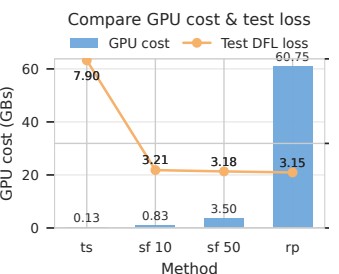

Figure 3: Learning curves for (a) score function with 10 and 50 samples (*sf 10* and *sf 50*) and reparameterization (*rp*), (b) score function and importance-weighted score function with 10 samples.

Figure 4: Computation cost vs. performance trade-off for diffusion DFL training

which then leads to unstable training. Intuitively, early diffusion steps (large noise levels) dominate the ELBO loss and its gradients, so if they happen to be sampled, they contribute disproportionately and noisily. With a small random subset of timesteps, the gradient estimate can thus be highly imbalanced and noisy, which causes training divergence in practice.

To address this, we adopt an importance sampling strategy for choosing diffusion timesteps. Empirically, as shown in **??** (b), the learning curves with the importance-weighted sampler are much smoother and more stable than with the uniform sampler. The score-function DFL training no longer diverges; instead, it converges cleanly, indicating that our variance reduction strategy successfully stabilizes the training process for diffusion DFL.

**Comparison on different problem sizes.** A key challenge for DFL is scalability: as the decision dimension grows, many methods degrade significantly Mandi et al. (2024). In this experiment, we investigate the performance of DFL methods under various decision dimensions in the stock portfolio. Specifically, we set the decision dimension range from 10 to 100 and report the test regrets. As summarized in Figure 5, the regret gap between diffusion-DFL and Gaussian and deterministic methods increases with the growth of dimensionality, which demonstrates that diffusion DFL scales effectively in more complex decision settings.

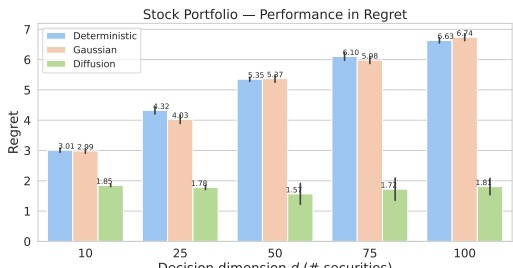

Figure 5: Test regret vs. decision dimension $d$ in the stock portfolio task.

# 8 CONCLUSION

We propose the first diffusion-based DFL approach for stochastic optimization, which trains a diffusion model to capture complex uncertainty in problem parameters. We develop two end-to-end training techniques to integrate the diffusion model into decision-making: reparameterization and score function approximation. As demonstrated with empirical evidence, the score function method drastically reduces memory and computation cost while achieving similar performance to reparameterization and being easy to train. Empirically, diffusion DFL achieves state-of-the-art results on multiple benchmarks, consistently outperforming both traditional two-stage methods and prior DFL approaches.

## REPRODUCIBILITY STATEMENT

We release an open-source repository containing all code, configuration files, and scripts needed to reproduce our results, including data generation and figure plotting. All proofs for the main paper are stated in the appendix with explanations and proper assumptions.

## ACKNOWLEDGMENTS

This work was supported by NSF grants (IIS-2403240, CCF-2326609, CNS-2146814, CPS-2136197, CNS-2106403, NGSDI-2105648); Schmidt Sciences AI2050 Fellowship; NIH grant R01HL184139;

gifts from Amazon, OpenAI, and Latitude AI; a Quad Fellowship; and the Caltech Resnick Sustainability Institute.

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

# A APPENDIX

**Notation.** We let $\frac{\partial f}{\partial x}$ denote the Jacobian matrix where $\left(\frac{\partial f}{\partial x}\right)_{i,j} := \frac{\partial f_i}{\partial x_j}$ and $\nabla_x f := \left(\frac{\partial f}{\partial x}\right)^\top$ denote the gradient. For a vector $v$, $D(v)$ denotes a diagonal matrix with $v$ on its diagonal. Let $P(\cdot)$ denote a probability distribution and $p(\cdot)$ denote a probability density; in particular, for diffusion models we use $P_\theta$ for the model's output distribution and $p_\theta$ for transition densities.

In this appendix, we derive the decision optimization problem with **general convex constraints** rather than merely linear constraints. Assume the optimization problem is

$$z_\theta^*(x) = \arg\min_z \mathbb{E}_{y \sim P_\theta(\cdot|x)}[f(y,z)], \quad \text{s.t. } h(x,z) \le 0, \ g(x,z) = 0,$$

where $h(x,z) \le 0$ denotes the convex inequalities constraints and $g(x,z) = 0$ denotes the equality constraints.

## A.1 PROOFS FOR SECTION 4

**Proposition A.1** (Reparameterization trick in diffusion models). *Let $T \in \mathbb{N}^+$, and suppose the reverse diffusion model defines a Gaussian distribution in Eq. 2 with fixed scalars $\sigma_t \ge 0$ and a standard normal prior $y_T \sim \mathcal{N}(0, I)$. Let $\{\epsilon_t\}_{t=0}^T$ be i.i.d. $\mathcal{N}(0, I)$. Then the model output $y$ can be expressed as a transformation $y = R(\epsilon_{0:T}, \theta \mid x)$ of a base noise distribution $\epsilon \sim P(\epsilon)$, where $R$ is differentiable in $\theta$. Also assume $\mathbb{E}_{y \sim P_\theta(\cdot|x)}[f(y,z)]$ is continuously differentiable. Then we have*

$$\nabla_\theta \mathbb{E}_{y \sim P_\theta(\cdot|x)}[f(y,z)] = \mathbb{E}_{\epsilon \sim P(\epsilon)} \left[ \left( \sum_{s=1}^T \left( \prod_{u=1}^{s-1} J_u \right) A_s \right)^\top \nabla_y f(R(\epsilon, \theta \mid x), z) \right],$$

*where $A_t := \frac{\partial \mu_\theta(y_t, t, x)}{\partial \theta}$, $J_t := \frac{\partial \mu_\theta(y_t, t, x)}{\partial y_t}$, and we define $\prod_{u=1}^0 J_u := I$.*

*Proof.* The conditional diffusion reverse process is defined as

$$y_{t-1} = \mu_\theta(y_t, t, x) + \sigma_t \epsilon_{t-1}, \qquad y_T = \epsilon_T,$$

where the noise term $\sigma_t \epsilon_{t-1}$ is $\theta$-independent. Differentiating both sides w.r.t. $\theta$ gives

$$\frac{\partial y_{t-1}}{\partial \theta} = \frac{\partial \mu_\theta(y_t, t, x)}{\partial \theta} + \frac{\partial \mu_\theta(y_t, t, x)}{\partial y_t} \frac{\partial y_t}{\partial \theta}.$$

Denote

$$A_t := \frac{\partial \mu_\theta(y_t, t, x)}{\partial \theta}, \quad J_t := \frac{\partial \mu_\theta(y_t, t, x)}{\partial y_t}, \quad G_t := \frac{\partial y_t}{\partial \theta}.$$

Thus, we have

$$G_{t-1} = A_t + J_t G_t, \quad G_T = 0.$$

Our final goal is:

$$\begin{aligned}
\nabla_\theta R(\epsilon_{0:T}, \theta|x) &= \frac{\partial y_0}{\partial \theta} = G_0 \\
&= A_1 + J_1 A_2 + J_1 J_2 A_3 + \cdots + J_1 \cdots J_{T_1} A_t \\
&= \sum_{s=1}^T \left( \prod_{u=1}^{s-1} J_u \right) A_s,
\end{aligned}$$

where we define $\prod_{u=1}^0 J_u := I$. Then, we have

$$\begin{aligned}
\nabla_\theta \mathbb{E}_{y \sim P_\theta(\cdot|x)}[f(y,z)] &= \nabla_\theta \mathbb{E}_{\epsilon \sim P(\epsilon)}[f(R(\epsilon, \theta|x), z)] \\
&= \mathbb{E}_{\epsilon \sim P(\epsilon)}[\nabla_\theta f(R(\epsilon, \theta|x), z)] \\
&= \mathbb{E}_{\epsilon \sim P(\epsilon)} \left[ \nabla_\theta R(\epsilon, \theta \mid x)^\top \nabla_y f(R(\epsilon, \theta|x), z) \right] \\
&= \mathbb{E}_{\epsilon \sim P(\epsilon)} \left[ \left( \sum_{s=1}^T \left( \prod_{u=1}^{s-1} J_u \right) A_s \right)^\top \nabla_y f(R(\epsilon, \theta|x), z) \right]
\end{aligned}$$

$\square$

**Lemma A.2** (Gradient of Reparameterization method)**.** *Assume the model prediction $y$ can be expressed as a transformation $y = T(\epsilon, \theta \mid x)$, $\epsilon \sim P(\epsilon)$. The total derivative of the decision objective $F$ w.r.t. $\theta$ can be computed as*

$$\frac{dF}{d\theta} = - \begin{bmatrix} \frac{dF}{dz^*} \\ 0 \\ 0 \end{bmatrix}^\top \begin{bmatrix} H & G^\top & Q^\top \\ D(\lambda^*)G & D(h(x, z^*)) & 0 \\ Q & 0 & 0 \end{bmatrix}^{-1} \begin{bmatrix} \mathbb{E}_{\epsilon \sim P(\epsilon)}[(\nabla_\theta T(\epsilon, \theta | x))^\top \nabla^2_{zy} f(z^*, y)] \\ 0 \\ 0 \end{bmatrix},$$

*where $H = \mathbb{E}_{y \sim P_\theta(\cdot|x)}[\nabla^2_{zz} f(y, z^*)] + \nabla^2_{zz}(\lambda^{*\top} h(x, z^*))$ is the Hessian of the Lagrangian with respect to $z$, $G = \nabla_z h(x, z^*)$ is the Jacobian of the inequality constraints in $z^*$, and $Q = \nabla_z g(x, z^*)$ is the Jacobian of the equality constraints in $z^*$.*

*Proof.* At the primal-dual optimal solution $(z^*_\theta, \lambda^*_\theta, \nu^*_\theta)$ to Eq. 1, the following KKT conditions must hold:

$$\nabla_z \mathcal{L}(\theta, z^*_\theta, \lambda_\theta, \nu_\theta; x) = 0,$$
$$\lambda_\theta \odot h(x, z^*_\theta) = 0,$$
$$g(x, z^*_\theta) = 0$$
$$\lambda_\theta \geq 0, \nu_\theta \geq 0,$$
$$h(x, z^*_\theta) \leq 0.$$

Since $h$ does not depend on $\theta$ here, we can apply Proposition A.1 to the KKT conditions to get

$$\frac{\partial \nabla_z \mathcal{L}}{\partial \theta} + \frac{\partial \nabla_z \mathcal{L}}{\partial z} \frac{\partial z^*}{\partial \theta} + \frac{\partial \nabla_z \mathcal{L}}{\partial \lambda^*} \frac{\partial \lambda^*}{\partial \theta} + \frac{\partial \nabla_z \mathcal{L}}{\partial \nu^*} \frac{\partial \nu^*}{\partial \theta}$$

$$= \mathbb{E}_{\epsilon \sim P(\epsilon)}[(\nabla_\theta T(\epsilon, \theta | x))^\top \nabla_y (\nabla_z f(z^*, y))] + \big( \mathbb{E}_{y \sim P_\theta(\cdot|x)}[\nabla^2_{zz} f(z^*, y)] + \nabla^2_{zz} h(x, z^*) \big) \frac{\partial z^*}{\partial \theta}$$

$$+ \nabla_z h(x, z^*) \frac{\partial \lambda^*}{\partial \theta} + \nabla_z g(x, z^*) \frac{\partial \lambda^*}{\partial \theta} = 0.$$

$$\frac{\partial \lambda^* \odot h(x, z^*)}{\partial z^*} \frac{\partial z^*}{\partial \theta} + \frac{\partial \lambda^* \odot h(x, z^*)}{\partial \lambda^*} \frac{\partial \lambda^*}{\partial \theta} = D(\lambda^*) \nabla_z h(x, z^*) \frac{\partial z^*}{\partial \theta} + D(h(x, z^*)) \frac{\partial \lambda^*}{\partial \theta} = 0.$$

In matrix form, we have

$$\begin{bmatrix} H & G^\top & Q^\top \\ D(\lambda^*)G & D(h(x, z^*)) & 0 \\ Q & 0 & 0 \end{bmatrix} \begin{bmatrix} \frac{\partial z^*}{\partial \theta} \\ \frac{\partial \lambda^*}{\partial \theta} \\ \frac{\partial \nu^*}{\partial \theta} \end{bmatrix} = - \begin{bmatrix} \mathbb{E}_{\epsilon \sim P(\epsilon)}[(\nabla_\theta T(\epsilon, \theta | x))^\top \nabla^2_{zy} f(z^*, y)] \\ 0 \\ 0 \end{bmatrix},$$

where $H = \mathbb{E}_{y \sim P_\theta(\cdot|x)}[\nabla^2_{zz} f(z^*, y)] + \nabla^2_{zz}(\lambda^{*\top} h(x, z^*)) + \nabla^2_{zz}(\nu^{*\top} g(x, z^*))$, $G = \nabla_z h(x, z^*)$, and $Q = \nabla_z g(x, z^*)$. Furthermore, if equalities and inequalities are affine (as in main paper), $H$ reduces to $\mathbb{E}_{y \sim P_\theta(\cdot|x)}[\nabla^2_{zz} f(y, z^*)]$ since $\nabla^2_{zz} h = \nabla^2_{zz} g = 0$.

By chain rule, we have

$$\frac{dF}{d\theta} = \begin{bmatrix} \frac{dF}{dz^*} \\ 0 \end{bmatrix}^\top \begin{bmatrix} \frac{\partial z^*}{\partial \theta} \\ \frac{\partial \lambda^*}{\partial \theta} \end{bmatrix}$$

$$= - \begin{bmatrix} \frac{dF}{dz^*} \\ 0 \\ 0 \end{bmatrix}^\top \begin{bmatrix} H & G^\top & Q^\top \\ D(\lambda^*)G & D(h(x, z^*)) & 0 \\ Q & 0 & 0 \end{bmatrix}^{-1} \begin{bmatrix} \mathbb{E}_{\epsilon \sim P(\epsilon)}[(\nabla_\theta T(\epsilon, \theta | x))^\top \nabla^2_{zy} f(z^*, y)] \\ 0 \\ 0 \end{bmatrix}.$$

$\square$

## A.2 Proofs for Section 5

**Proposition A.3.** *Let $P_\theta(y \mid x)$ be a probability density parameterized by $\theta \in \Theta$, and let $f : \mathcal{Y} \times \mathbb{R}^d \to \mathbb{R}$ be a scalar-valued function that does not depend on $\theta$. Fix any $z \in \mathbb{R}^d$. Suppose that there exists some neighborhood $N(\theta_0) \subseteq \Theta$ around $\theta_0 \in \Theta$ such that the following 3 assumptions are satisfied:*

1. For all $\theta \in N(\theta_0)$, the function $h(y) := P_\theta(y \mid x) f(y, z)$ is integrable;
2. For all $\theta \in N(\theta_0)$ and almost all $y \in \mathcal{Y}$, the gradient $\nabla_\theta P_\theta(y \mid x)$ exists; and
3. There exists an integrable function $g : \mathcal{Y} \to \mathbb{R}$ that dominates $\nabla_\theta P_\theta(y \mid x)$. That is, for all $\theta \in N(\theta_0)$ and almost all $y \in \mathcal{Y}$, $\|\nabla_\theta P_\theta(y \mid x)\|_1 \le |g(y)|$.

*Then,*

$$\nabla_\theta \mathbb{E}_{y \sim P_\theta(\cdot \mid x)}[f(y, z)] = \mathbb{E}_{y \sim P_{\theta_0}(\cdot \mid x)}[f(y, z) \cdot \nabla_\theta \log P_{\theta_0}(y \mid x)].$$

*Proof.* We make use of the log-derivative trick:

$$P_{\theta_0}(y \mid x) \cdot \nabla_\theta \log P_{\theta_0}(y \mid x) = \frac{P_{\theta_0}(y \mid x)}{P_{\theta_0}(y \mid x)} \cdot \nabla_\theta P_{\theta_0}(y \mid x) = \nabla_\theta P_{\theta_0}(y \mid x).$$

Then

$$\nabla_\theta \, \mathbb{E}_{y \sim P_{\theta_0}(\cdot \mid x)}[f(y, z)] = \nabla_\theta \int_\mathcal{Y} f(y, z) P_{\theta_0}(y \mid x) \, \mathrm{d}y$$

$$= \int_\mathcal{Y} \nabla_\theta \left[ f(y, z) \, P_{\theta_0}(y \mid x) \right] \mathrm{d}y \qquad \text{(by Leibniz integral rule)}$$

$$= \int_\mathcal{Y} f(y, z) \, P_{\theta_0}(y \mid x) \, \nabla_\theta \log P_{\theta_0}(y \mid x) \, \mathrm{d}y \qquad \text{(by log-derivative trick)}$$

$$= \mathbb{E}_{y \sim P_{\theta_0}(\cdot \mid x)} \left[ f(y, z) \nabla_\theta \log P_{\theta_0}(y \mid x) \right].$$

$\square$

**Lemma A.4** (Gradient of Score Function). *The total derivative of the decision objective $F$ w.r.t. $\theta$ can be computed as*

$$\frac{dF}{d\theta} = - \begin{bmatrix} \frac{dF}{dz^*} \\ 0 \\ 0 \end{bmatrix}^\top \begin{bmatrix} H & G^\top & Q^\top \\ D(\lambda^*)G & D(h(x, z^*)) & 0 \\ Q & 0 & 0 \end{bmatrix}^{-1} \begin{bmatrix} \mathbb{E}_{y \sim P_\theta(\cdot \mid x)}[\nabla_z f(z^*, y)(\frac{dELBO}{d\theta})^\top] \\ 0 \\ 0 \end{bmatrix}.$$

*Proof.* Differentiate this KKT system w.r.t. $\theta$ and applying Proposition A.3 yields

$$\frac{\partial \nabla_z \mathcal{L}}{\partial \theta} + \frac{\partial \nabla_z \mathcal{L}}{\partial z} \frac{\partial z^*}{\partial \theta} + \frac{\partial \nabla_z \mathcal{L}}{\partial \lambda^*} \frac{\partial \lambda^*}{\partial \theta} + \frac{\partial \nabla_z \mathcal{L}}{\partial \nu^*} \frac{\partial \nu^*}{\partial \theta}$$

$$= \mathbb{E}_{y \sim P_\theta(\cdot \mid x)}[\nabla_z f(z^*, y)(\nabla_\theta \log P_\theta(y \mid x))^\top] + (\mathbb{E}_{y \sim P_\theta(\cdot \mid x)}[\nabla_{zz}^2 f(z^*, y)] + \nabla_{zz}^2(\lambda^{\star \top} h(x, z^*))) \frac{\partial z^*}{\partial \theta}$$

$$+ \nabla_z h(x, z^*) \frac{\partial \lambda^*}{\partial \theta} + \nabla_z g(x, z^*) \frac{\partial \nu^*}{\partial \theta} = 0.$$

$$\frac{\partial \lambda^* \odot h(x, z^*)}{\partial z^*} \frac{\partial z^*}{\partial \theta} + \frac{\partial \lambda^* \odot h(x, z^*)}{\partial \lambda^*} \frac{\partial \lambda^*}{\partial \theta} = D(\lambda^*)\nabla_z h(x, z^*)\frac{\partial z^*}{\partial \theta} + D(h(x, z^*))\frac{\partial \lambda^*}{\partial \theta} = 0.$$

In matrix form, this becomes

$$\begin{bmatrix} H & G^\top & Q^\top \\ D(\lambda^*)G & D(h(x, z^*)) & 0 \\ Q & 0 & 0 \end{bmatrix} \begin{bmatrix} \frac{\partial z^*}{\partial \theta} \\ \frac{\partial \lambda^*}{\partial \theta} \\ \frac{\partial \nu^*}{\partial \theta} \end{bmatrix} = - \begin{bmatrix} \mathbb{E}_y[\nabla_z f(z^*, y)(\nabla_\theta \log P_\theta(y \mid x))^\top] \\ 0 \\ 0 \end{bmatrix}, \qquad (11)$$

where $H = \mathbb{E}_{y \sim P_\theta(\cdot \mid x)}[\nabla_{zz}^2 f(z^*, y)] + \nabla_{zz}^2(\lambda^{* \top} h(x, z^*))$, $G = \nabla_z h(x, z^*)$.

Applying the chain rule to $F$ now gives

$$\frac{dF}{d\theta} = \begin{bmatrix} \frac{dF}{dz^*} \\ 0 \\ 0 \end{bmatrix}^\top \begin{bmatrix} \frac{\partial z^*}{\partial \theta} \\ \frac{\partial \lambda^*}{\partial \theta} \\ \frac{\partial \nu^*}{\partial \theta} \end{bmatrix}$$

$$= - \begin{bmatrix} \frac{dF}{dz^*} \\ 0 \\ 0 \end{bmatrix}^\top \begin{bmatrix} H & G^\top & Q^\top \\ D(\lambda^*)G & D(h(x, z^*)) & 0 \\ Q & 0 & 0 \end{bmatrix}^{-1} \begin{bmatrix} \mathbb{E}_{y \sim P_\theta(\cdot \mid x)}[\nabla_z f(z^*, y)(\nabla_\theta \log P_\theta(y \mid x))^\top] \\ 0 \\ 0 \end{bmatrix}.$$

Then, we replace $\nabla_\theta \log P_\theta(y \mid x)$ with the gradient of ELBO score for sample $y$ and have

$$\frac{dF}{d\theta} = - \begin{bmatrix} \frac{dF}{dz^*} \\ 0 \\ 0 \end{bmatrix}^\top \begin{bmatrix} H & G^\top & Q^\top \\ D(\lambda^*)G & D(h(x,z^*)) & 0 \\ Q & 0 & 0 \end{bmatrix}^{-1} \begin{bmatrix} \mathbb{E}_{y \sim P_\theta(\cdot|x)}[\nabla_z f(z^*,y) \cdot (\nabla_\theta \mathrm{ELBO}(\theta;y))^\top] \\ 0 \\ 0 \end{bmatrix}.$$

$\square$

**Remark A.5** (Why we cannot compute the gradient using score-matching). *One may attempt to apply the chain rule $\nabla_\theta \log P_\theta(y|x) = \nabla_\theta y \nabla_y \log P_\theta(y|x)$, and then estimate $\nabla_y \log P_\theta(y_{t+1}|x) \approx \nabla_y \log P_\theta(y_{t+1}|y_t,x) \approx s_\theta(y_t,t,x)$ via score-matching (Song et al., 2021) (using the learned score $s_\theta(y_t,t,x)$ of the diffusion model). However, this approach is invalid in our setting: Under the log-trick, $y$ is treated as a free variable and $\theta$ enters only through $P_\theta(y|x)$, so the pathwise term $\nabla_\theta y$ does not exist (see Appendix A.2 for derivation). Our ELBO-based surrogate (Eq. (8)) avoids this obstacle entirely.*

### A.3 PROOF FOR EQ. 10

Based on the results in Lemma A.4, we have

$$\frac{dF}{d\theta} = - \begin{bmatrix} \frac{dF}{dz^*} \\ 0 \\ 0 \end{bmatrix}^\top \begin{bmatrix} H & G^\top & Q^\top \\ D(\lambda^*)G & D(h(x,z^*)) & 0 \\ Q & 0 & 0 \end{bmatrix}^{-1} \begin{bmatrix} \mathbb{E}_{y \sim P_\theta(\cdot|x)}[\nabla_z f(z^*,y) \cdot (\nabla_\theta \mathrm{ELBO}(\theta;y))^\top] \\ 0 \\ 0 \end{bmatrix}$$

$$= - \underbrace{\begin{bmatrix} \frac{dF}{dz^*} \\ 0 \\ 0 \end{bmatrix}^\top \begin{bmatrix} H & G^\top & Q^\top \\ D(\lambda^*)G & D(h(x,z^*)) & 0 \\ Q & 0 & 0 \end{bmatrix}^{-1}}_{:=u(\theta)^\top} \frac{d}{d\theta} \begin{bmatrix} \mathbb{E}_{y \sim P_\theta(\cdot|x)}[\nabla_z f(z^*,y)\,\mathrm{ELBO}] \\ 0 \\ 0 \end{bmatrix}$$

$$= \frac{d}{d\theta} \mathbb{E}_{y \sim P_\theta(\cdot|x)}[u(\theta)^\top \underbrace{\begin{bmatrix} [\nabla_z f(z^*,y)] \\ 0 \\ 0 \end{bmatrix} \mathrm{ELBO}]}_{:=w_\theta(y)}$$

$$= \frac{d}{d\theta} \mathbb{E}_{y \sim P_\theta(\cdot|x)}[\mathrm{detach}[w_\theta(y)]\mathrm{ELBO}].$$

### A.4 GRADIENT ERROR OF ELBO VS. LOG-LIKELIHOOD

Under mild assumptions, we can prove an upper bound for the error of our ELBO gradient approximation. Recall from Eq. 7, we define ELBO as a lower bound for log-likelihood. Here, we can actually write an equality relationship between them:

$$\log P_\theta(y_0) = \mathrm{ELBO}(y_0;\theta) + \mathrm{KL}(q(z)||p_\theta(z|y_0)),$$

where $z$ denotes a latent variable and $\mathrm{ELBO}(y_0,\theta) := \mathbb{E}_{q(z)}[\log p_\theta(x,z)] - \mathbb{E}_{q(z)}[\log q(z)]$. Let denote the score function as $s_\theta(z;y_0) := \nabla_\theta \log p_\theta(y_0,z)$. We immediately have the following two results:

$$\nabla_\theta \log P_\theta(y_0) = \nabla_\theta \log \int p_\theta(y_0,z)\,dz$$

$$= \frac{1}{p_\theta(y_0)} \int \nabla_\theta p_\theta(y_0,z)\,dz$$

$$= \frac{1}{p_\theta(y_0)} \int p_\theta(y_0,z)\nabla_\theta \log p_\theta(y_0,z)\,dz$$

$$= \int \nabla_\theta \log p_\theta(y_0,z)\frac{p_\theta(y_0,z)}{p_\theta(y_0)}\,dz$$

$$= \mathbb{E}_{p_\theta(z|y_0)}[\nabla_\theta \log p_\theta(y_0,z)]$$

$$= \mathbb{E}_{p_\theta(z|y_0)}[s_\theta(z;y_0)].$$

Similarly,

$$\nabla_\theta \mathrm{ELBO}(y_0; \theta) = \nabla_\theta \mathbb{E}_{q(z)}[\log p_\theta(y_0, z)]$$
$$= \mathbb{E}_{q(z)}[\nabla_\theta \log p_\theta(y_0, z)]$$
$$= \mathbb{E}_{q(z)}[s_\theta(z; y_0)].$$

Then we are ready to state the following proposition:

**Proposition A.6** (ELBO gradient approximation error). *Let $s_\theta(z; y_0) := \nabla_\theta \log p_\theta(y_0, z)$ denote the joint score function. If $\sup_w \|s_\theta(z; y_0)\| \leq B(\theta, y_0)$ for some finite upper bound $B(\theta, y_0) > 0$, then*

$$\|\nabla_\theta \log P_\theta(y_0) - \nabla_\theta \mathrm{ELBO}(y_0; \theta)\| \leq \sqrt{2} B(\theta, y_0) \sqrt{\mathrm{KL}(q(y_{1:T} \mid y_0) \| p_\theta(y_{1:T} \mid y_0))}.$$

*Proof.*

$$\|\nabla_\theta \log P_\theta(y_0) - \nabla_\theta \mathrm{ELBO}(y_0; \theta)\| = \left\| \mathbb{E}_{p_\theta(z|y_0)}[s_\theta(z; y_0)] - \mathbb{E}_{q(z)}[s_\theta(z; y_0)] \right\|$$
$$= \left\| \int s_\theta(z; y_0)(p_\theta(z \mid y_0) - q(z)) \, dz \right\|$$
$$\leq \int \|s_\theta(z; y_0)(p_\theta(z \mid y_0) - q(z))\| \, dz$$
$$= \int \|s_\theta(z; y_0)\| \, |p_\theta(z \mid y_0) - q(z)| \, dz$$
$$\leq B(\theta, y_0) \int |p_\theta(z \mid y_0) - q(z)| \, dz$$
$$= B(\theta, y_0) \cdot 2\mathrm{TV}(q(z), p_\theta(z \mid y_0))$$
$$\leq \sqrt{2} B(\theta, y_0) \sqrt{\mathrm{KL}(q(z) \| p_\theta(z \mid y_0))},$$

where TV denotes the total variation distance of probability measures: $\mathrm{TV}(p, q) = \frac{1}{2} \int |q(x) - p(x)| dx$, and the last inequality is due to $\mathrm{TV}(q, p) \leq \sqrt{\frac{1}{2}\mathrm{KL}(q\|p)}$. $\qquad\square$

In the context of the diffusion model, the latent variable $z = y_{1:T}$ is the diffusion trajectory, $q(y_{1:T})$ is the distribution over the diffusion trajectory $y_{1:T}$, and $p_\theta(y_{1:T}|y_0)$ is the corresponding diffusion reverse process. Thus, whenever the variational approximation is good (small KL), the ELBO gradient is provably close to the true score.

## A.5 EMPIRICAL EVIDENCE FOR ELBO GRADIENT APPROXIMATION

Assume our model is $\theta = (A, B, c)$, and the noise is predicted by $\epsilon_\theta(y_t, t, x) = A_t y_t + B_t x + c_t$. True data $y \sim \mathcal{N}(Wx, I)$. In this way, there is a closed-form solution for true $\nabla_\theta \log p(y_0|x)$ since $y_0$ is a Gaussian and $y_t = \sqrt{\bar\alpha_t} y_0 + \sqrt{1 - \bar\alpha_t}\epsilon$.

## A.6 DETERMINISTIC OPTIMIZATION AND GAUSSIAN MODEL IN STOCHASTIC OPTIMIZATION

**Deterministic Optimization**  Since deterministic can also be viewed as a reparameterization trick without any randomness $\epsilon$, we can reuse our derivation in Section 4 and compute the gradient $\nabla_x F(x)$ by Eq. 5.

**Gaussian Model in Stochastic Optimization**  Since there are many recent papers that use the Gaussian model as a predictor for stochastic DFL, we also claim that it has the reparameterization and score function form.

1. Reparameterization with Gaussian Model. Also using the reparameterization trick:

$$\nabla_\theta \mathbb{E}_{y \sim P_\theta(\cdot|x)}[f(y, z)] = \mathbb{E}_{\epsilon \sim P(\epsilon)}[(\nabla_\theta y^\top \nabla_y f(y, z)].$$

But with the predictor instantiated as a Gaussian model, i.e., $y$ is computed by $y = R(\epsilon, \theta|x))^\top$ and $R$ is a Gaussian model.

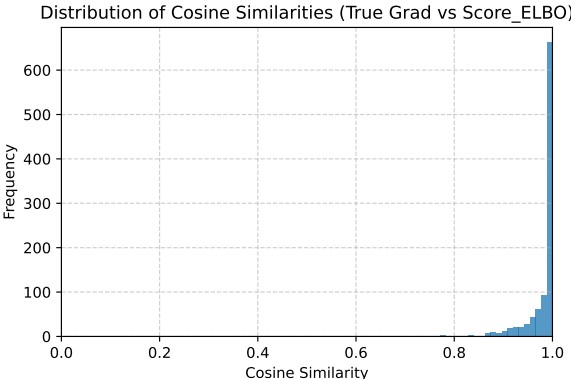

Figure 6: Cosine similarity between the true gradient and the estimated gradient using a linear model.

2. Score Function with Gaussian Model. Recall we need to approximate the term $\log P_\theta(y|x)$ for the diffusion model. However, this term has a closed-form for a Gaussian model. Assume our Gaussian model is $\mathcal{N}(\mu_\theta, \Sigma_\theta)$, then the negative log-likelihood is

$$\log P_\theta(y \mid x) = -\frac{1}{2}[(y - \mu_\theta)^\top \Sigma_\theta^{-1}(y - \mu_\theta) + \log \det \Sigma_\theta + d \log(2\pi)].$$

Then, the gradient for the score function can be calculated using $\nabla_\theta \log P_\theta(y \mid x)$. Gaussian models are powerful tools for many DFL tasks. However, we want to claim that the diffusion model is more general and requires less model tuning and model assumptions.

### A.7 CONNECTIONS WITH OFFLINE CONTEXTUAL BANDIT METHODS

DFL and offline contextual bandits both involve learning from contextual features to make decisions, but they differ in key assumptions and objectives. Specifically, in DFL, the decision $z$ is a solution of solving a known optimization problem, with explicit constraints and objective (e.g., quadratic objective with convex constraints). In other words, DFL explicitly incorporates the known structure of the decision task into the learning problem. Predict-then-optimize methods specifically aim to leverage this structure of the optimization problem to learn better predictions under fewer samples. This incorporation of task structure is the key motivation for predict-then-optimize methods.

In an offline bandit view, each decision (action) $z$ is obtained by a policy that is learned directly without using the fact that $z$ comes from solving a particular optimization problem. In theory, if the model is expressive enough, an offline bandit algorithm can learn the decision with enough samples, but it may ignore the underlying optimization structure and thus require more samples.

Here, we implement an offline full-information (the true cost function is revealed) contextual bandit method using policy-based learning methods. Specifically, we first sample a batch $B$ from the offline dataset, and then compute the decisions (actions) by a policy network $\phi$. Our goal is to train the policy network $\phi$ to minimize the following loss function:

$$L(\phi) = \frac{1}{|B|} \sum_{(x,y) \in B} f(\phi(x), y).$$

After training $T$ epochs, we evaluate $\phi$ in a test set and report the results.

### A.8 TRAINING DETAILS AND HYPERPARAMETERS

We summarize our model settings for the deterministic DFL, Gaussian DFL, and diffusion DFL in Table 2.

For all experiments, we perform 10 random seeds to evaluate variability.

**Implementation Details** Stochastic optimization introduces two main computational bottlenecks: (i) sampling multiple Monte Carlo instances from the diffusion model, and (ii) repeatedly solving the downstream optimization problem. We address both in our implementation:

| Parameter | Deterministic MLP | Gaussian MLP | Diffusion Model |
|---|---|---|---|
| **Model Architecture** | | | |
| Trunk (layers × width) | $2 \times 1024$ | $2 \times 1024$ | $2 \times 1024$ |
| Activation | ReLU / SiLU | ReLU / SiLU | SiLU |
| Inputs | $x \in \mathbb{R}^{d_x}$ | $x \in \mathbb{R}^{d_x}$ | $[y_t,\ t,\ x]$ |
| Time embedding | — | — | Sinusoidal $t$ (16-d) $\to$ 2 FC + SiLU |
| Output head | $\hat{y} \in \mathbb{R}^{d_y}$ | $\mu(x),\ \log \sigma^2(x)$ | $\hat{\epsilon}_\theta(y_t, t, x)$ |
| Uncertainty | None (point) | Gaussian $\mathcal{N}(\mu, \mathrm{diag}(\sigma^2))$ | Diffusion process $P_\theta(y \mid x)$ |
| **Training** | | | |
| Loss | MSE$(y)$ | Gaussian NLL | Weighted denoising MSE |
| DFL gradient | Implicit diff. via KKT | (1) Reparam (Gaussian) (2) Gaussian score | (1) Reparam (Diffusion) (2) Weighted-ELBO score |
| Sample size $M$ (synthetic/power/portfolio) | — | 10/25/50 | 10/25/50 |
| Learning rate | $1 \times 10^{-4}$ (typical) | Reparam: $1 \times 10^{-5}$ Score-fn: $8 \times 10^{-6}$ | Reparam: $1 \times 10^{-5}$ Score-fn: $8 \times 10^{-6}$ |
| **Inference** | | | |
| Procedure | Use $\hat{y}$ directly | Sample $M$ outputs $y^{(m)}$ | Reverse diffusion to get $M$ samples $y^{(m)}$ |

Table 2: Architectural and training differences among deterministic, Gaussian, and diffusion-based DFL methods.

- **Parallel diffusion sampling.** We sample in parallel across many noise realizations, which makes efficient use of hardware and reduces the per-sample overhead. In our supplementary material, for each task (`power_sched`, `stock_portfolio`, `synthetic_example`), the file `{task}/diffusion_opt.py` implements the function `sample_elbo`, which accepts batched inputs and runs the diffusion sampler for the entire batch in parallel. This allows us to draw many Monte Carlo samples in *one single forward pass*.
- **Parallel optimization solving.** The downstream optimization solver is also executed in parallel across different samples. In `{task}/cvxpy_{task}.py`, the function `cvxpy_{task}_parallel_kkt` constructs a batched KKT system for the Monte Carlo samples and *solves these batched KKT systems in parallel* (across CVXPY worker jobs).

## A.9 DETAILS OF SYNTHETIC TASK

In this example, we consider a factory that decides how much to manufacture for each of $d \in \mathbb{N}$ products. The parameter $Y \in \mathbb{R}^d$ represents the *profit margin* for each product, i.e., $Y_i$ is the profit per unit of product $i$; due to uncertainty in market conditions, $Y$ is uncertain. The factory's decision $z \in [0, C]^d$ represents how much of each product to manufacture, where $C$ is the maximum capacity for each product. For simplicity, we do not consider any contextual features $x$ in this example. That means DFL learns a distribution that generates $y$ that can minimize the decision objective.

Suppose that the factory has a risk-averse cost function $f(y, z) = \exp(-y^\top z)$, which indicates that the factory wants to put a larger weight on the product with higher profit $Y_i$. Intuitively, if the factory knew $Y$ exactly, then the optimal strategy would be all-or-nothing: set $z_i = C$ if $Y_i > 0$, or $z_i = 0$ if $Y_i < 0$. Likewise, with respect to a point prediction of $Y$, the optimal deterministic decision $z_{\det}^* \in \{0, C\}^d$ is attained on the boundary of the feasible set.

Under uncertainty, the decision-maker seeks to minimize the **expected cost** by solving a stochastic optimization problem:

$$z_{\text{sto}}^* \in \arg \min_{z \in [0, C]^d} \mathbb{E}_{y \sim P_\theta(\cdot | x)}[\exp(-y^\top z)].$$

In this stochastic case, the optimal investment $z_{\text{sto}}^*$ typically lies in the interior of the feasible region, which balances the potential high reward of investing against the risk of losses.

Then, we compute the necessities for diffusion DFL:

$$H = \mathbb{E}_{y \sim P_\theta(\cdot | x)}[\nabla_{zz}^2 g(z^*, y)] + (\lambda^*)^\top \nabla_{zz}^2 h(x, z^*) = \exp(-y^\top z) y y^\top$$

$$G = \nabla_z h(x, z^*) = -\exp(-y^\top z) y.$$

For reparameterization, we have

$$(\frac{dloss}{d\theta})^\top = \begin{bmatrix} \frac{dloss}{dz^*} \\ 0 \end{bmatrix}^\top \begin{bmatrix} \frac{\partial z^*}{\partial \theta} \\ \frac{\partial \lambda^*}{\partial \theta} \end{bmatrix}$$

$$= -\begin{bmatrix} \frac{dloss}{dz^*} \\ 0 \end{bmatrix}^\top \begin{bmatrix} H & G^\top \\ D(\lambda^*)G & D(h(x, z^*)) \end{bmatrix}^{-1} \begin{bmatrix} \frac{1}{M} \sum_{i=1}^M (\nabla_\theta y_i)^\top \nabla_{zy}^2 g(z^*, y_i) \\ 0 \end{bmatrix}$$

where $\nabla_{zy}^2 g(z^*, y) = \exp(-y^\top z)(yz^\top - I_d)$ in this case.

For the score function, we have

$$(\frac{dloss}{d\theta})^\top = -\begin{bmatrix} \frac{dloss}{dz^*} \\ 0 \end{bmatrix}^\top \begin{bmatrix} H & G^\top \\ D(\lambda^*)G & D(h(x, z^*)) \end{bmatrix}^{-1} \begin{bmatrix} \mathbb{E}_y[\nabla_z g(z^*, y)(\frac{dELBO}{d\theta})^\top] \\ 0 \end{bmatrix}$$

$$\approx -\begin{bmatrix} \frac{dloss}{dz^*} \\ 0 \end{bmatrix}^\top \begin{bmatrix} H & G^\top \\ D(\lambda^*)G & D(h(x, z^*)) \end{bmatrix}^{-1} \begin{bmatrix} \frac{1}{M} \sum_{i=1}^M \nabla_z g(z^*, y_i)(\frac{dELBO_i}{d\theta})^\top \\ 0 \end{bmatrix}.$$

## A.10 DETAILS ON POWER SCHEDULE TASK

This task involves a 24-hour electricity generation scheduling problem with uncertain demand. The decision $z \in \mathbb{R}^{24}$ represents the electricity output to schedule for each hour of the next day. The uncertainty $y \in \mathbb{R}^{24}$ represents the actual power demand for each of the 24 hours. The goal is to meet demand as closely as possible at minimum cost. We also consider a decision cost function that penalizes storage, excess generation, and ramping following Donti et al. (2017):

1. Let $\gamma_s$ and $\gamma_a$ be the per-unit costs of shortage (not meeting demand) and excess (over-generation), respectively. We use $\gamma_s = 50$ and $\gamma_e = 0.5$ in our experiment.

2. Let $c_r$ be a penalty on hour-to-hour changes in generation. We use $c_r = 0.4$ in appropriate units.

Formally, if $z = (z_1, \ldots, z_{24})$ and $y = (y_1, \ldots, y_{24})$, the loss for a single day is

$$\min_z \mathbb{E}_{y \sim P_\theta(\cdot|x)}[f(y, z)] = \sum_{i=1}^{24} \mathbb{E}_{y \sim P_\theta(\cdot|x)}[\gamma_s[y_i - z_i]_+ + \gamma_e[z_i - y_i]_+ + \frac{1}{2}(z_i - y_i)^2]$$

s.t. $|z_i - z_{i-1}| \leq c_r$ for all $i \in \{1, 2, \ldots, 24\}$.

Then, we compute the necessities for diffusion DFL:

$$H = \mathbb{E}_{y \sim P_\theta(\cdot|x)}[\nabla^2_{zz} g(z^*, y)] + (\lambda^*)^\top \nabla^2_{zz} h(x, z^*) = I_n$$

$$G = \nabla_z h(x, z^*) = \begin{bmatrix} -1 & 1 & 0 & \cdots & 0 \\ 0 & -1 & 1 & \cdots & 0 \\ \vdots & & \ddots & \ddots & \vdots \\ 0 & \cdots & 0 & -1 & 1 \end{bmatrix}.$$

For reparameterization, we have

$$(\frac{dloss}{d\theta})^\top = \begin{bmatrix} \frac{dloss}{dz^*} \\ 0 \end{bmatrix}^\top \begin{bmatrix} \frac{\partial z^*}{\partial \theta} \\ \frac{\partial \lambda^*}{\partial \theta} \end{bmatrix}$$

$$= -\begin{bmatrix} \frac{dloss}{dz^*} \\ 0 \end{bmatrix}^\top \begin{bmatrix} H & G^\top \\ D(\lambda^*)G & D(h(x, z^*)) \end{bmatrix}^{-1} \begin{bmatrix} \frac{1}{M} \sum_{i=1}^{M} (\nabla_\theta y_i)^\top \nabla^2_{zy} g(z^*, y_i) \\ 0 \end{bmatrix}$$

where $\nabla^2_{zy} g(z^*, y) = -I$ in this case.

For the score function, we have

$$(\frac{dloss}{d\theta})^\top = -\begin{bmatrix} \frac{dloss}{dz^*} \\ 0 \end{bmatrix}^\top \begin{bmatrix} H & G^\top \\ D(\lambda^*)G & D(h(x, z^*)) \end{bmatrix}^{-1} \begin{bmatrix} \mathbb{E}_y[\nabla_z g(z^*, y)(\frac{dELBO}{d\theta})^\top] \\ 0 \end{bmatrix}$$

$$\approx -\begin{bmatrix} \frac{dloss}{dz^*} \\ 0 \end{bmatrix}^\top \begin{bmatrix} H & G^\top \\ D(\lambda^*)G & D(h(x, z^*)) \end{bmatrix}^{-1} \begin{bmatrix} \frac{1}{M} \sum_{i=1}^{M} \nabla_z g(z^*, y_i)(\frac{dELBO_i}{d\theta})^\top \\ 0 \end{bmatrix}.$$

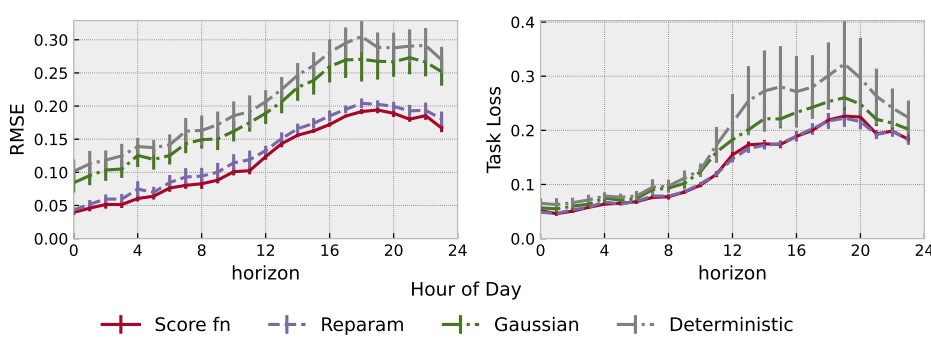

Figure 7: Results on the 24-hour power grid scheduling task.

**Dataset.** We use real, public historical electricity load data from the PJM regional grid (PJM Interconnection, 2025). The features $x$ for each day include: the previous day's 24-hour load profile, the previous day's temperature profile, calendar features, and seasonal sinusoidal features. In total, $d_x = 150$ features for each day were constructed. We normalized all input features for training. The target label $y$ is the next day's 24-hour load vector.

For completeness, we include an extended comparison of different sample sizes in Figure 9, which further highlights that additional samples yield diminishing returns in accuracy while linearly increasing compute cost. We also find that adding a small regularizer during DFL training can help the model learning the data distribution and avoid some bad local minima, leading to a stable training process.

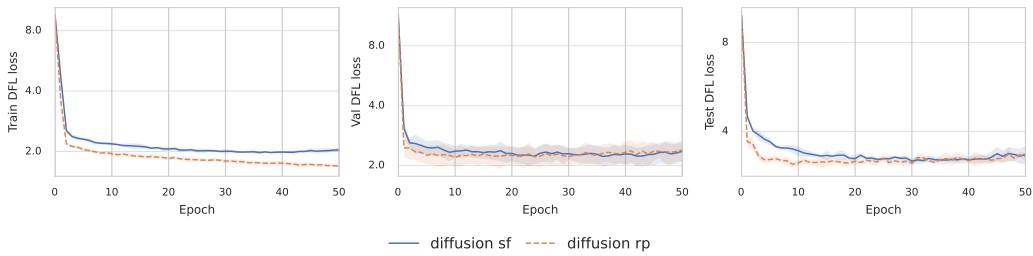

Figure 8: Train, validation, and test DFL loss for diffusion DFL using score function vs. reparameterization. Solid lines show the mean loss over multiple runs with different random seeds; shaded regions indicate standard deviation across runs.

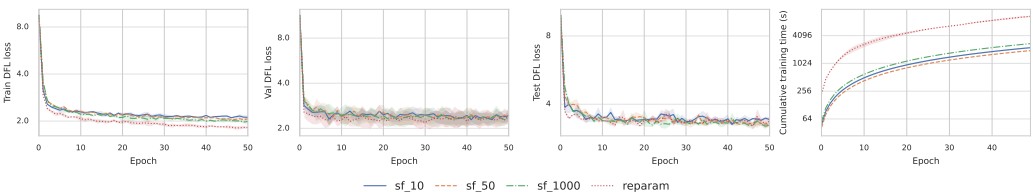

Figure 9: Comparison between different sample sizes for score function and reparameterization.

## A.11 DETAILS ON STOCK PORTFOLIO TASK

We consider a mean-variance portfolio optimization problem with uncertain returns. Here $y \in \mathbb{R}^n$ represents the random next-day returns of $n$ assets (stocks), and $z \in \mathbb{R}^n$ are the portfolio weights we assign to each asset (the fraction of our capital invested in each stock). Our goal is to maximize expected return while keeping the risk (variance) low. This can be written as minimizing a loss that is a negative expected return plus a quadratic penalty on variance:

$$\min_z \ \mathbb{E}_{y \sim P_\theta(\cdot|x)}[f(y,z)] = \mathbb{E}_{y \sim P_\theta(\cdot|x)} \left[ \frac{\alpha}{2} z^\top y y^\top z - y^\top z \right], \quad \text{s.t.} \quad z^\top \mathbf{1} = 1, \ 0 \le z_i \le 1.$$

Then, we compute the necessities for diffusion DFL:

$$H = \mathbb{E}_{y \sim P_\theta(\cdot|x)}[\nabla^2_{zz} f(z^*, y)] + (\lambda^*)^\top \nabla^2_{zz} h(x, z^*) + \nabla^2_{zz}(\nu^{*\top} g(x, z^*)) = \alpha \, \mathbb{E}_{y \sim P_\theta(y|x)}[y y^\top],$$

$$G = \nabla_z h(x, z^*) = \begin{bmatrix} I_n, \\ -I_n \end{bmatrix}$$

$$Q = \nabla_z g(x, z^*) = \mathbf{1}^\top.$$

For reparameterization, we have

$$\left(\frac{dloss}{d\theta}\right)^\top = \begin{bmatrix} \frac{dloss}{dz^*} \\ 0 \\ 0 \end{bmatrix}^\top \begin{bmatrix} \frac{\partial z^*}{\partial \theta} \\ \frac{\partial \lambda^*}{\partial \theta} \\ \frac{\partial \nu^*}{\partial \theta} \end{bmatrix}$$

$$\approx - \begin{bmatrix} \frac{dloss}{dz^*} \\ 0 \\ 0 \end{bmatrix}^\top \begin{bmatrix} H & G^\top & Q^\top \\ D(\lambda^*)G & D(h(x, z^*)) & 0 \\ Q & 0 & 0 \end{bmatrix}^{-1} \begin{bmatrix} \frac{1}{M} \sum_{i=1}^M (\nabla_\theta y_i)^\top \nabla^2_{zy} g(z^*, y_i) \\ 0 \\ 0 \end{bmatrix}$$

where $\nabla^2_{zy} f(z^*, y) = \mathbb{E}_y[2\alpha y^\top z - 1]$ in this case.

For the score function, we have

$$\left(\frac{dloss}{d\theta}\right)^\top = \begin{bmatrix} \frac{dloss}{dz^*} \\ 0 \\ 0 \end{bmatrix}^\top \begin{bmatrix} H & G^\top & Q^\top \\ D(\lambda^*)G & D(h(x, z^*)) & 0 \\ Q & 0 & 0 \end{bmatrix}^{-1} \begin{bmatrix} \mathbb{E}_y[\nabla_z g(z^*, y)(\frac{dELBO}{d\theta})^\top] \\ 0 \\ 0 \end{bmatrix}$$

$$\approx - \begin{bmatrix} \frac{dloss}{dz^*} \\ 0 \\ 0 \end{bmatrix}^\top \begin{bmatrix} H & G^\top & Q^\top \\ D(\lambda^*)G & D(h(x, z^*)) & 0 \\ Q & 0 & 0 \end{bmatrix}^{-1} \begin{bmatrix} \frac{1}{M} \sum_{i=1}^M \nabla_z g(z^*, y_i)(\frac{dELBO_i}{d\theta})^\top \\ 0 \\ 0 \end{bmatrix},$$

where $\nabla_z g(z^*, y_i) = \alpha y_i y_i^\top z^* - y_i$.

**Dataset.** We use daily stock prices from 2004–2017 for constituents of the S&P 500 index (Quandl WIKI dataset, 2025). We obtained this data via Quandl's API (specifically WIKI pricing data; the user will need a Quandl API key to replicate. We compute daily returns for each stock (percentage change). To construct features $x$, we use a rolling window of recent history for each asset. Specifically, for each day, a data point is for predicting next day's returns: we include the past 5 days of returns for each of the $n$ assets, past 5 days of trading volume for each asset, plus some aggregate features. To avoid an explosion of dimension with large $n$, we also include PCA-compressed features: we take the top principal components of the last 5-day return matrix to summarize cross-asset trends. In the end, for $n = 50$ assets, we ended up with $d_x = 28$ features. All features are normalized and we use a time-series split: first 70% of days for training (2004–2013), next 10% for validation (2014), last 20% for test (2015–2017). We evaluate performance on the test set by simulating the portfolio selection every day and computing the average return achieved.

### A.12 (ADDITIONAL TASK) DETAILS ON INVENTORY STOCK PROBLEM

We also validate our approaches on a toy inventory control problem. In this task, the uncertain demand $y$ is drawn from a multi-modal distribution (a mixture of Gaussians), where we vary the number of mixture components $K$ to control the distribution complexity:

$$p(x) = \sum_{j=1}^{K} \pi_j \phi(x; \mu_j, \Sigma_j),$$

where $\pi_j$ is the probability of choosing component j and $\phi(x; \mu_j, \Sigma_j)$ is a multivariate Gaussian density with parameter $(\mu_j, \Sigma_j)$. The cost function follows the standard newsvendor formulation with piecewise penalties for under-stock and over-stock:

$$f(y, z) = c_0 z + \frac{1}{2} q_0 z^2 + c_b[y - z]_+ + \frac{1}{3} r_b([y - z]_+^3) + c_h[z - y]_+ + \frac{1}{3} r_h([z - y]_+^3).$$

Our learning objective is to minimize the expected cost over this stochastic demand, i.e., a stochastic optimization problem:

$$\min_z L(\theta) = \mathbb{E}_{y \sim P(\cdot|x)}[f(y, z)] \quad \text{s.t. } 0 \leq z \leq z_{max}.$$

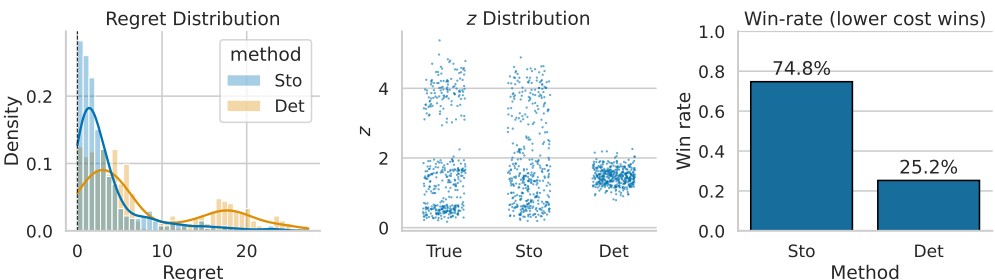

Figure 10: Toy decision task comparing deterministic and diffusion DFL. *Left*: distribution of per-instance regret (lower is better). *Middle*: distribution of chosen decision $z$ in the lower-level; the stochastic method tracks the true distribution $z^*$ more closely. *Right*: pairwise win-rate on test set; a large fraction of costs from the stochastic method are lower than the deterministic one, indicating that modeling uncertainty yields better decisions.

We compare a deterministic DFL model against our diffusion DFL model on this toy task. Figure 10 summarizes the results, where the diffusion model (stochastic DFL) achieves substantially lower regret on average than the deterministic model. Besides, we observe that the decision $z$ obtained by our diffusion method closely tracks the true optimal decisions $z^*$ by capturing the multi-modal demand uncertainty, whereas the deterministic predictor's decisions deviate more. In Figure 10 (c), we directly compare the decision outcomes via a win-rate: the fraction of test instances where one method achieves a lower cost than the other. The diffusion DFL method attains a win-rate of about 75% against the deterministic baseline, which confirms that modeling uncertainty leads to better downstream decisions

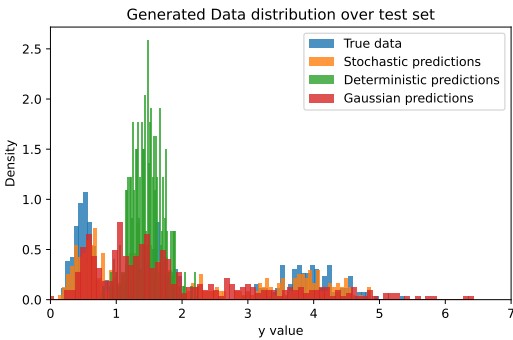

Figure 11: Prediction distribution for inventory stock problem.

### A.13 ADDITIONAL RELATED WORKS AND DISCUSSIONS

**Stochastic optimization**    Making decisions under uncertainty is a classic topic in operations research and machine learning (Shalev-Shwartz et al., 2009). Stochastic optimization formulations explicitly consider uncertainty by optimizing the expected objective over a distribution of unknown parameters. A common approach is the Sample Average Approximation (SAA) (Kleywegt et al., 2002; Arjevani et al., 2020; Wang et al., 2024b), which draws many samples from the estimated distribution and solves an approximated deterministic problem minimizing the average cost. While SAA can handle arbitrary uncertainty distributions in theory, it becomes very computationally expensive and still does not consider the distribution during optimization (Kim et al., 2015). It will lead to optimizing the *sample mean*, which may yield a decision that performs poorly if reality often falls into one of several distinct models far from the mean (Kim et al., 2015; Elmachtoub & Grigas, 2022).

When is Diffusion-DFL useful? Our experiments suggest that Diffusion DFL will be especially useful in decision-making settings where the uncertain parameter in the objective is high-dimensional, has a multimodal distribution, and limited training data is available. We believe that promising directions of future work include accommodating uncertainty in constraints in addition to the objective function, investigating the trade-off between model calibration and decision-making quality, and adapting Diffusion DFL to online decision-making settings.

**Offline Contextual Bandits (OCB)**    Offline contextual bandit approaches provide an alternative way to learn decisions from contextual features by directly optimizing a policy that maps each context to an action (Agarwal et al., 2020; Nguyen-Tang et al., 2021; Brandfonbrener et al., 2021; Gabbianelli et al., 2024; Sakhi et al., 2023). A key difficulty is that feedback is typically *partial*: the log reveals rewards only for actions taken by a behavior policy, so learning and evaluation rely on off-policy estimators and assumptions such as overlap between the learned and behavior policies. Recent OCB methods address this challenge via optimistic or pessimistic objectives (Nguyen-Tang et al., 2021; Wang et al., 2024a), variance control (Li et al., 2011; 2012; Wang et al., 2017), conservative regularization (Swaminathan & Joachims, 2015), and robust optimization (Yang et al., 2023) to avoid out-of-distribution actions. In all these methods, the policy is learned purely from logged data, without using the fact that the decision comes from solving a particular optimization problem. This is fundamentally different from DFL in terms of the prior knowledge and optimization structure. In DFL, the decision $z$ is a solution of solving a known optimization problem, with explicit constraints and objectives. In this paper, we empirically show that Diffusion-DFL achieves better performance than the policy-based OCB method.

