# OpenReview forum: "Diffusion-DFL: Decision-focused Diffusion Models for Stochastic Optimization"
_ICLR.cc/2026/Conference — ICLR 2026 Poster_

### Official Review · Reviewer_1MXz · 2025-10-27

**Soundness:** 3
**Presentation:** 3
**Contribution:** 3
**Rating:** 6
**Confidence:** 2

**Summary:**

The paper introduces a novel DFL approach that leverages diffusion models to represent the distribution of uncertain parameters. To mitigate the issue raised by using diffusion models, i.e., sequential sampling procedure, the authors develop two algorithms: reparametrization and score function. Experiment results demonstrate that using diffusion models outperforms prior baselines.

**Strengths:**

- Utilizing diffusion models in decision-focused learning is sound and novel
- Authors successfully mitigate issues raised when using diffusion models in decision-focused learning
- Experiments and further analysis are comprehensive

**Weaknesses:**

- The authors say one benefit of utilizing diffusion models is capturing a multi-modal distribution. Does the task provided benefit when we capture the multi-modal distribution? It would be nice to persuade readers why we should capture multi-modal distribution in decision-focused learning.

**Questions:**

- I'm not familiar with decision-focused learning. Are there any other benchmarks for evaluating the performance of decision-focused learning? It seems that the three optimization tasks provided in the paper are not complex enough.

---

> ### Author Response · Authors · 2025-11-20
> **Response to Reviewer 1Mxz**
>
> We thank the reviewer for thoughtful and constructive feedback and comments. Please see our rebuttal summary, which highlights our key responses to reviewer feedback as well as revisions to our paper. Below, we provide a point-by-point response to reviewer 1MXz's review.
>
> >Q1: The authors say one benefit of utilizing diffusion models is capturing a multi-modal distribution. Does the task provided benefit when we capture the multi-modal distribution? It would be nice to persuade readers why we should capture multi-modal distribution in decision-focused learning.
>
> In many real-world settings such as modeling stock portfolio returns and electricity demand, the data distribution is naturally **multi-modal** (see Figure 9 from Liao et al.; Figure 6 from Backer et al.). Furthermore, previous works have shown that modeling the full predictive distribution for decision-making is in general advantageous to compared to a single point estimate. Only under limited conditions on the optimization problem can a point estimate recover the optimal decision (Schutte et al, 2025; Homem-de-Mello et al, 2024).
>
> Therefore, using a stochastic predictor such as a diffusion model, which naturally captures multi-modality and uncertainty, can significantly improve performance in DFL.
>
> Reference:
> (We also thank Reviewer dJh7 for pointing out the first reference that our paper misses.)
> - Liao et al. Modeling and Analysis of Residential Electricity Consumption Statistics: A Tracy-Widom Mixture Density Approximation. IEEE Access, 2020.
> - Stijn De Backer et al. Characterizing asymmetric and bimodal long-term financial return distributions through quantum walks. arXiv:2505.13019, 2025.
> - Schutte et al. Sufficient Decision Proxies for Decision-Focused Learning. arXiv:2505.03953, 2025.
> - Homem-de-Mello, et al. Forecasting Outside the Box: Application-Driven Optimal Pointwise Forecasts for Stochastic Optimization. arXiv:2411.03520, 2024.
>
> >Q2: Are there any other benchmarks for evaluating the performance of decision-focused learning?
>
> The two real-world problems we consider (power grid (Donti et al., 2017) and stock portfolio (Wang et al., 2020)) are standard and representative in the DFL literature. We agree that expanding to more domains is an important future direction, and we plan to explore more diverse settings in follow-up work.
>
> Reference:
> - Donti et al. Task-based End-to-end Model Learning in Stochastic Optimization. NIPS 2017.
> - Wang et al. Automatically Learning Compact Quality-aware Surrogates for Optimization Problems. NeurIPS 2020.

---

> > ### Comment · Reviewer_1MXz · 2025-11-26
> >
> > Thanks for the response. At least for me, introducing diffusion models into decision-focused learning is novel. Therefore, I keep my positive score.

---

### Official Review · Reviewer_ZKgV · 2025-10-27

**Soundness:** 3
**Presentation:** 3
**Contribution:** 3
**Rating:** 6
**Confidence:** 3

**Summary:**

This paper introduces Diffusion-DFL, a first framework that integrates diffusion probabilistic models into decision-focused learning (DFL) — an approach that trains predictive models to directly optimize downstream decision quality rather than prediction accuracy.
Unlike prior DFL methods, which rely on deterministic or simple probabilistic predictors (e.g., Gaussian), Diffusion-DFL uses a conditional diffusion model to capture potentially complex and multimodal uncertainty in the parameters of a downstream stochastic optimization problem. The decision layer then optimizes the expected cost under this learned distribution.

To train the model end-to-end, the authors develop two gradient estimators:
1) Reparameterization estimator: backpropagates through the diffusion sampling process, providing an exact but memory-intensive gradient.
2) Score-function estimator: avoids backpropagating through diffusion by applying the log-trick, estimating gradients via the model’s ELBO  — drastically reducing memory and compute cost with minimal accuracy loss.

Experiments on synthetic manufacturing, power scheduling, and portfolio optimization tasks show that Diffusion-DFL consistently yields better decision outcomes than both two-stage and deterministic DFL baselines, while the score-function version achieves similar performance to reparameterization with over 400× lower memory usage.

**Strengths:**

The paper introduces the first integration of diffusion models into the decision-focused learning framework, allowing richer uncertainty modeling than deterministic approaches.

- The proposed score-function estimator is a practical contribution: it replaces costly pathwise differentiation with an ELBO-based surrogate, maintaining decision performance while reducing memory and compute by several orders of magnitude.
- The methodology is well-grounded theoretically (via implicit differentiation and the log-trick) and empirically validated on multiple domains, showing consistent improvements in decision quality.
- Clear ablation studies (memory vs. accuracy, variance reduction, decision dimension scaling) demonstrate that the approach is both effective and scalable.

**Weaknesses:**

- The score-function approximation relies on an implicit assumption of local smoothness between the model’s likelihood and decision-loss landscapes, which is not formally analyzed or guaranteed, although supported empirically.
- Experimental settings are limited to convex or low-dimensional decision problems; performance on more complex or nonconvex tasks (where diffusion expressiveness would matter most) remains unclear.

**Questions:**

The score-function estimator seems to rely on local smoothness between the diffusion model’s likelihood landscape and the downstream decision cost; can the authors discuss or empirically validate when this assumption might fail, e.g., in highly nonconvex or discontinuous decision problems? Have the authors tested Diffusion-DFL in nonconvex or combinatorial settings, where diffusion’s multimodality could be beneficial?

---

> ### Author Response · Authors · 2025-11-20
> **Response to Reviewer ZKgV**
>
> We thank the reviewer for thoughtful and constructive feedback and comments. Please see our rebuttal summary, which highlights our key responses to reviewer feedback as well as revisions to our paper. Below, we provide a point-by-point response to reviewer ZKgV's review.
>
> >Q1:The score-function approximation relies on an implicit assumption of local smoothness between the model’s likelihood and decision-loss landscapes, which is not formally analyzed or guaranteed, although supported empirically.
>
> Thank you for your suggestion. We additionally complement our empirical evidence with a new theoretical result in our revised manuscript to address your concern. In Proposition A.7 of the revised appendix, we prove an upper bound on the norm of the difference between the true log-likelihood gradient and the ELBO gradient: under mild assumptions,
> $$
> ||\nabla_\theta \log P_\theta(y_0) - \nabla_\theta \text{ELBO}(y_0;\theta)|| \leq \sqrt{2}B(\theta, y_0) \sqrt{\text{KL}(q(y_{1:T} \mid y_0)\|\|p_\theta(y_{1:T}\mid y_0))},
> $$
> where $B(\theta, y_0)$ is a problem-dependent constant. Thus, whenever the variational approximation is good (small KL), the ELBO gradient is provably close to the true score. Together with our experiments showing high cosine similarity between the two gradients, this provides both theoretical and empirical support for using the ELBO gradient as a surrogate to the score.
>
> Besides, our method is related in spirit to prior works in varitional inference. In VAEs (Kingma & Welling, 2014; Rezende & Mohamed, 2015; Burda et al., 2015), the exact log-likelihood gradients are often intractable in practice, where  tractable surrogate losses like ELBO are commonly used to compute the gradient information.
>
> We thank the reviewer for the insightful feedback and will revise the main paper accordingly to include this new result.
>
> References:
> - Diederik P Kingma, Max Welling. Auto-Encoding Variational Bayes. ICLR, 2014.
> - Rezende, Danilo, and Shakir Mohamed. Variational inference with normalizing flows. ICML, 2015.
> - Burda, Yuri, Roger Grosse, and Ruslan Salakhutdinov. Importance weighted autoencoders. arXiv:1509.00519, 2015.
>
> >Q2: The score-function estimator seems to rely on local smoothness between the diffusion model’s likelihood landscape and the downstream decision cost; can the authors discuss or empirically validate when this assumption might fail, e.g., in highly nonconvex or discontinuous decision problems?
> Q3: Experimental settings are limited to convex or low-dimensional decision problems; performance on more complex or nonconvex tasks (where diffusion expressiveness would matter most) remains unclear. Will the diffusion’s multimodality could be beneficial in non-convex or combinatorial settings?
>
> Our score function surrogate does **not** depend on the convexity of the optimization problems. To see this, our score function actually comes from the derivation Eq. 34, which holds without any convexity assumption on $f$ or on downstream optimization problem.
>
> Our current experiments focus on convex or tractable problems mainly for clarity and ease of comparison.  In principle, our method can also be applied when the decision layer is non-convex, as long as we can (approximately) solve the corresponding KKT system or obtain useful gradients through the solver. We agree that this is exactly the regime where the extra expressiveness and multi-modality of diffusion models might matter even more.
>
> Reference:
> - Bryan Wilder, Bistra Dilkina, Milind Tambe. Melding the Data-Decisions Pipeline: Decision-Focused Learning for Combinatorial Optimization. AAAI, 2019.

---

### Official Review · Reviewer_dJh7 · 2025-11-01

**Soundness:** 3
**Presentation:** 4
**Contribution:** 3
**Rating:** 8
**Confidence:** 5

**Summary:**

This paper presents a Decision Focused Learning method where the output of the estimator is stochastic, rather than being a point estimate, and the downstream problem is also a stochastic optimization one. As a main advantage, the approach can better address decision problems where no deterministic formulation can result in the correct optimal solution under uncertainty, e.g. because any individual realization leads to an either maximizing or minimizing a variables, while the best value under under uncertainty would be somewhat in the middle.

The authors use a diffusion process to generate their estimate and present two techniques to enable differentiation. The first -- technically more precise, but expensive in terms of computation time and memory -- is based on a reparameterization of each diffusion step and implicit differentiation. The second -- much lighter from a computational standpoint -- is based on Score Function Gradient Estimation and an Evidence Lower Bound. Using a diffusion process enables modeling complex distributions

The approach is tested on selected benchmarks from the literature, and at least in one realistic (but still far from true practical applicability) problem it achieved a very significant advantage in terms of solution quality.

**Strengths:**

The insight about the limited expressiveness (for lack of a better term) of deterministic DFL is correct and definitely useful, even if not entirely novel.

The empirical results on the stock portfolio problem provide evidence that a combination of stochastic prediction and a complex distribution can improve the effectiveness of the DFL approach very significantly, beyond what is possible for either technique. Remarkably, this is true even when the two-phase approach is rather expressive and should be capable of approximating well the ground truth parameter distribution (the PFL diffusion method).

The suggested differentiation method based on SFGE + ELBO appears to be rather good at improving the computational efficienty, with no major drop in terms of solution quality.

The SFGE/ELBO method also seems to be applicable, with a few minor modifications, to a _much_ more general setting than the one presented in eq. (2). Since the method only requires function evaluation, it seems it could easily account for solution repairs costs (similar to the other SFGE method by Silvestri et al.) and therefore, combinatorial problems, two-stage problems, or decision problems with uncertain parameters in the constraints.

**Weaknesses:**

It is my opinion that the paper strengths outweigh its weaknesses. That said, I can find some, of which a few are major and a few minor.

The first major weakness I can find is that the approach requires solving a stochastic optimization problem per example, and per epoch. Stochastic optimization problems can be very challenging to address and the corresponding solution methods are well known for their rather poor scalability. Combined with the already computational cost of DFL training, this significantly reduces the practical appeal of the method. That said, given the improvements reported in one benchmark, I think there are definitely practical settings with relatively lightweight optimization problems that can benefit from the proposed technique.

The second major weakness is again related to applicability. If enough data is available and a sufficiently expressive (distributional) ML model is used, the approximation error for the ground truth distribution will eventually converge to zero. Under these conditions, a Prediction Focused Learning (i.e. two phase) pipeline should be capable of reaching optimal or close to optimal results, similarly to what happens in a classical DFL setup with non-misspecified ML models. The empirical results corroborate this intuition, with the Diffusion TS method reaching high quality results in the synthetic benchmark. It would seem that the ideal application case for the proposed method is one where a deterministic decision problem is inadequate, a complex distribution is needed, and the problem cost is not well aligned with the likelihood function used in PFL. While situations like this certainly exists, the applicability of the method is also reduced.

Here is a list of some minor weaknesses:

* The fact that deterministic DFL may not be expressive enough for some optimization problems under uncertainty is also established in [1]. Given that this paper is very recent and that the expressiveness claim is not at the center of this work, this issue is definitely minor. That said, it might be worth including a citation
* The Gaussian method by Silvestri et al. is not stochastic, but deterministic: the Gaussian distribution is used just for smoothing, while the optimization problem works with a single parameter vector (both at training and inference time). By reading Appendix A.5, it seems the authors instead implemented an actual Gaussian stochastic DFL pipeline, based on their own techniques, but using a simpler distribution. This is again a minor issue, since the original method by Silvestri et al. would likely work even worse in this setting (though it would be much more scalable)
* This might be my fault, but I did not easily find the number of samples used in the empirical evaluation

[1] Schutte, Noah, et al. "Sufficient Decision Proxies for Decision-Focused Learning." arXiv preprint arXiv:2505.03953 (2025).

**Questions:**

* Can you confirm that the Gaussin DFL method is in fact a stochastic method, i.e. using a stochastic optimization problem with multiple samples downstream?
* How many samples were used in the empirical evaluation?

---

> ### Author Response · Authors · 2025-11-20
> **Response to Reviewer dJh7**
>
> We thank the reviewer for thoughtful and constructive feedback and comments. Please see our rebuttal summary, which highlights our key responses to reviewer feedback as well as revisions to our paper. Below, we provide a point-by-point response to reviewer dJh7's review.
>
> >Q1: Computation cost is high for solving stochastic optimization.
>
> Indeed, stochastic optimization introduces two main computational bottlenecks: (i) sampling multiple Monte Carlo instances from diffusion, and (ii) solving downstream optimization repeatedly. We address both in our implementation by batched parallel diffusion sampling and batched KKT solving. A section is added to explain this in Appendix A.8 of our revised manuscript.
>
> >Q2: Two-stage methods can achieve optimal solution when enough data is available and a sufficiently expressive (distributional) ML model is used. The applicability of the method is reduced.
>
> We agree with the reviewer that two-stage methods can achieve optimality when the predictive model is sufficiently expressive and sufficient data is available to fully capture the distribution. We have added a discussion paragraph to clarify this point (see Seciton 8 of revised manuscript). As highlighted there, Diffusion-DFL is most beneficial in the complementary regime: when the decision problem is sensitive to uncertainty, or when the available dataset is too limited to accurately represent the full distribution.
>
> >Q3 (Minor): Missing reference for Schutte, Noah, et al.
>
> Thank you for pointing us to the Schutte et al.'s work. We have added this reference.
>
> >Q4 (Minor): The Gaussian method by Silvestri et al. is not stochastic, but deterministic.
>
> Correct. The Gaussian method by Silvestri et al. does not consider a "real" stochastic optimization problem. They first sample $\hat y$ from a stochastic model and then repeatedly perform a *deterministic optimization* to get $z^*(\hat y)$ for every sample. In contrast, we actually draw **multiple samples** from the stochastic predictor, then solve **one** stochastic optimization problem and differentiate through it. Besides, due to the need to solve the optimization problem multiple times, Silvestri et al.'s method has significantly more computation cost than ours.
>
> >Q5 (Minor): How many samples were used in the empirical evaluation?
>
> We apologize for the omission. The sample sizes for stochastic optimization are 10 for the synthetic task, 25 for the power scheduling task, and 50 for the stock portfolio task.

---

> ### Comment · Reviewer_dJh7 · 2025-11-20
> **My opinion remains positive**
>
> As stated in the title, my opinion of the paper is still positive. I think the work makes some valuable contribution in the context of DFL research.
>
> Concerning the current response, I think it's worth distinguishing between training time and inference time scalability.
>
> From this point of view, it's important to observe that all deterministic DFL methods are strictly (and possibly much) more scalable than the proposed approach, which requires multiple samples also at inference time. The trade-off is that, as clearly shown in the results, for some settings using a deterministic optimization problems is not sufficiently expressive and the solution quality can be much worse.
>
> In terms of training time scalability, the situation is murkier. Assuming the same number of samples is used and/or generated from the distributional predictor:
>
> 1. The proposed DFL method baed on stochastic optimization needs to solve a single problem with multiple samples. It pays less overhead, but needs to solve a more complex problem.
> 2. Deterministic DFL methods need to solve multiple optimization problems, one per sample. They pay more overhead, but each problem they solve is smaller
>
> On an NP-hard optimization problems, option 2 will eventually be always better than option 1 as the problem scale grows.

---

> > ### Author Response · Authors · 2025-11-21
> > **Response to Reviewer dJh7**
> >
> > We thank the reviewer for the positive assessment.
> >
> > >Q1: It's worth distinguishing between training time and inference time scalability.
> >
> > We thank the reviewer for the helpful comment on scalability.
> >
> > We want to clarify that deterministic and stochastic DFL solve problems with the same decision variables and constraints; only objective changes (single sample vs. average over a fixed number of samples). So the dimensionality of the underlying problem and complexity remain mostly the same in both cases, up to constant factors. While we acknowledge that the empirical difficulty of solving an instance may depend on the specific form of the objective, this primarily affects constant factors rather than the asymptotic scaling in problem size.

---

### Official Review · Reviewer_Azep · 2025-11-04

**Soundness:** 3
**Presentation:** 4
**Contribution:** 1
**Rating:** 0
**Confidence:** 5

**Summary:**

This paper studies a paradigm termed decision-focused learning, where the goal is to solve a decision-making problem that depends on uncertain parameters that must be learned. The authors develop a computational pipeline to propagate uncertainty from the learning procedure to the decision procedure using diffusion models, and show that incorporating uncertainty results in improved decision-making performance.

**Strengths:**

I note two key strengths:
* **Writing attention to detail.** This paper is reasonably polished, and it is clear the authors know how to write a paper to an appropriate technical standard.
* **Reasonably-careful experiments.** The authors not only demonstrate good performance on a key benchmark of interest compared to pure-deterministic methods, they also include various ablation studies to check individual parts of their conclusions.

In an ordinary review, I would offer many more details on these points. However, in light of a very serious issue which I will bring to light shortly, I will omit these in the interest of time.

**Weaknesses:**

This paper has a critical weakness because of which I am confident it needs to be rejected: **a very significant portion of this work's findings are known and covered in standard textbooks, but are presented as novel.** In particular, the paper's actual novelty is limited to introducing diffusion models to well-known decision-making paradigms. If re-written completely in a manner that reflects this contribution, I do believe the authors have done sufficient development to warrant a paper, but the gap between the current manuscript and what would be appropriate is, to say it again, sufficient to warrant a complete rewrite. This is unfortunately far too much to address in the rebuttal phase.

Allow me to elaborate:
* **Decision-focused learning is presented as a novel emerging paradigm, but is not new, because the authors' formulation is equivalent to a (stochastic) contextual bandit.** Let me be completely precise: the authors define their setting via the optimization objective $\mathbb{E}f(y^\star, z^\star(x))$ where $(x,y^\star) \sim D$ and $z^\star$ is parameterized by some vector $\theta$ which is to be optimized. Let us define the notation $\phi^\star(z) = f(y^\star, z)$, so the objective becomes $\mathbb{E} \phi^\star(z^\star(x))$ where $(x,\phi^\star) \sim D$. With this notation change, it is clear that the authors' setting is a (stochastic) contextual bandit: we are given a random context vector x which is correlated with some random objective $\phi^\star$ (either in a known way in the Bayesian case, or unknown way in the non-Bayesian case), and our goal is to choose the optimal map from context to actions. With this notational link made, there is a substantial literature of prior work the authors could consider and compare with, which is not mentioned in this work. Note, in particular, that if D is assumed known, maximizing the objective is equivalent to minimizing regret, which is the usual performance objective considered because it behaves much better mathematically.
* **Using stochastic models inside dynamic programming is not new.** Indeed, this is covered in Chapter 1.5 of "Sequential Decision Analytics and Modeling" by Warren Powell, to name just one book that includes this topic.
* **In the contextual bandit literature, the need for uncertainty is well-understood.** This is the basis not only for classical upper confidence bound algorithms, but also methods based on approximate dynamic programming such as expected improvement algorithms from the Bayesian optimization literature.
* **Even if one is interested purely in practical aspects involving diffusion models, one should benchmark against standard algorithms.** For example, I see no reason why, in light of the above equivalence, any DFL algorithm in the authors' sense cannot directly be benchmarked against contextual kernel UCB.
* **Section 4 consists entirely of known ideas.** In particular, the application of the reparametrization trick to handle objectives defined as expectations of this kind is, at this point, completely standard and need not be introduced except in code. The discussion on differentiating through the solution of a parameterized optimization problem is also standard, though less than reparametrization tricks - this should be deferred to an appendix.
* **In Section 5, the key ideas involving score estimation are essentially the same as the standard REINFORCE trick from reinforcement learning. Moreover, the algorithmic justification for using ELBO as a surrogate to the score is not technically sound.** The authors' argument, essentially, is to state that $f(x) \leq g(x)$ on basis of an evidence lower bound, and argue that this implies $\nabla f(x) \approx \nabla g(x)$, which is not necessarily true even if $|f(x) - g(x)| \leq \epsilon$ for an arbitrary given $\epsilon$. Rather than relying on a theoretical explanation that does not follow, I believe the authors should instead lean on their empirical results in Figure 2, which shows these are close, as this avoids a misleading explanation.

Summarizing, I believe this work's only contribution which is novel is the exploration of diffusion models in the context of stochastic dynamic programming. This contribution is a reasonable fit for ICLR and could make for a good paper, but the paper needs to be re-written in a manner that reflects this.

**Questions:**

Have I missed some critical contribution in the above points that should justify a different conclusion?

---

> ### Author Response · Authors · 2025-11-20
> **Response to Reviewer Azep (pt 1/2)**
>
> We appreciate the reviewer for their valuable comments. But we believe some of reviewer’s concerns appear to stem from conflating our **offline** predict-then-optimize setting with the **online** stochastic contextual bandit setting. In the bandit setting, the learner can interact with the environment to learn and minimize the regret sequentially, while in our DFL problem, the learner cannot interact with the environment and has to construct actions from an offline dataset following the predict-then-optimize pipeline. We would like to highlight that the difference in offline vs. online makes DFL and bandits completely different problems with substantially different challenges.
>
> >Q1: Decision-focused learning (DFL) is presented as a novel emerging paradigm, but is not new, because the authors' formulation is equivalent to a (stochastic) contextual bandit.
>
> As we wrote above, we recognize that while DFL and stochastic contextual bandits are related, they are fundamentally different. The contextual bandit problem models a sequential decision-making process, whereas DFL considers the offline batch setting.
>
> We also wish to clarify that DFL is a popular direction in operations research to derive actions from offline datasets. Our main contribution in this paper is the introduction of stochastic objective to DFL and how to differentiate through it with diffusion models.
>
>
> > Q2: Using stochastic models inside dynamic programming is not new.
>
> Although stochastic models are common in the bandit literature, for the offline setting in our case, the use of **diffusion models** to formulate the stochastic optimization in the predict-then-optimize framework is new to the best of our knowledge.
>
> > Q3: In the contextual bandit literature, the need for uncertainty is well-understood.
>
> Although the use of uncertainty is common in the bandit literature, the use of diffusion models to quantify decision-focused uncertainty in the offline predict-then-optimize setting is new. In particular, we note that many recent DFL papers (Jeon et al., 2025; Yang et al., 2025) still use a deterministic predictor in uncertainty-aware tasks (like power scheduling). The use of uncertainty in bandits should not be used to discourage the use of uncertainty in another field.
>
> Reference:
> - Jeon et al. Locally Convex Global Loss Network for Decision-Focused Learning. AAAI, 2025.
> - Yang et al. DFF: Decision-Focused Fine-Tuning for Smarter Predict-Then-Optimize with Limited Data. AAAI, 2025.
>
> > Q4: Even if one is interested purely in practical aspects involving diffusion models, one should benchmark against standard algorithms.
>
> The stochastic contextual bandits are mainly about the **online learning** and regret minimization. In contrast, we consider the **offline** predict-then-optimize setting and thus **online** algorithms do not apply here. Due to this discrepancy, we mainly benchmark our algorithm with many other offline algorithms in the DFL/predict-then-optimize literature to compare the performance.
>
> > Q5: Section 4 consists entirely of known ideas.
>
> While the reparameterization trick is not new, our work is the first to apply it to diffusion models in the predict-then-optimize framework (as we describe in Section 4). Our contributions include
> - We are the first work to introduce diffusion models into DFL.
> - We show that naively using reparameterization in DFL works but it may suffer from *high computation and memory cost*.
> - We develop a lightweight score-function formulation to train the diffusion models efficiently in DFL and show superior performance in our experiments

---

> ### Author Response · Authors · 2025-11-20
> **Response to Reviewer Azep (pt 2/2)**
>
> > Q6: Moreover, the algorithmic justification for using ELBO as a surrogate to the score is not technically sound.
>
> Thank you for your suggestion. We additionally complement our empirical evidence with a new theoretical result in our revised manuscript to address your concern. In Proposition A.7 of the revised appendix, we prove an upper bound on the norm of the difference between the true log-likelihood gradient and the ELBO gradient: under mild assumptions,
> $$
> ||\nabla_\theta \log P_\theta(y_0) - \nabla_\theta \text{ELBO}(y_0;\theta)|| \leq \sqrt{2}B(\theta, y_0) \sqrt{\text{KL}(q(y_{1:T}\mid y_0)\|\|p_\theta(y_{1:T}\mid y_0))},
> $$
> where $B(\theta, y_0)$ is a problem-dependent constant. Thus, whenever the variational approximation is good (small KL), the ELBO gradient is provably close to the true score. Together with our experiments showing high cosine similarity between the two gradients, this provides both theoretical and empirical support for using the ELBO gradient as a surrogate to the score.
>
> Besides, our method is related in spirit to prior works in variational inference. In VAEs (Kingma & Welling, 2014; Rezende & Mohamed, 2015; Burda et al., 2015), the exact log-likelihood gradients are often intractable in practice, where tractable surrogate losses like ELBO are commonly used to compute the gradient information.
>
> We thank the reviewer for the insightful feedback and will revise the main paper accordingly to include this new result.
>
> References:
> - Diederik P Kingma, Max Welling. Auto-Encoding Variational Bayes. ICLR, 2014.
> - Rezende, Danilo, and Shakir Mohamed. Variational inference with normalizing flows. ICML, 2015.
> - Burda, Yuri, Roger Grosse, and Ruslan Salakhutdinov. Importance weighted autoencoders. arXiv:1509.00519, 2015.

---

> > ### Comment · Reviewer_Azep · 2025-11-20
> >
> > Thank you for your comments and for answering the questions. In light of these, let me provide some additional context for the area chair to consider my review.
> > * **Q1.** The authors' response emphasizes the key difference between their setting and the one I explicitly described in the notation I wrote in my response - namely, my setting is online, theirs is offline. By analogy, this is similar to the difference between online and offline reinforcement learning. **I agree with this characterization, but it does not change the gravity of my key concern about novelty and positioning**. There are existing papers on offline bandits, mirroring the literature on offline reinforcement learning, but **neither the word "bandit" nor the word "reinforcement" is mentioned in the paper - not even offline reinforcement learning in spite of its relevance**. Ideas from this literature need to be discussed and benchmarked against, at minimum so that readers know that prior work exists in this area.
> >
> > * **Q2.** Thank you, I agree. What I am asking is for you to re-write your paper to reflect this framing, and present your contributions in an honest and non-misleading way that mirrors your response - namely, that, as you say "the use of diffusion models to formulate the stochastic optimization in the predict-then-optimize framework is new". Your paper is not written in this manner, it needs to be re-written and re-submitted to reflect this.
> >
> > * **Q3.** Thank you for pointing out these papers. I have briefly looked at them, and it is my view that neither of them should have been published either without additional benchmarking and a modified framing that reflects the existence of prior work, for the same reasons I have given for your work. I have no idea what level of scrutiny these works received at AAAI since I have never published there, but given the stochasticity of ML reviewing it is likely that reviewers there simply missed this critical issue.
> >
> > * **Q4.** See Q1: there is an existing literature on offline learning in bandits, and more generally on offline reinforcement learning. Your paper makes no mention of this.
> >
> > * **Q5.** I would agree that, in the strict sense, this is new. However, it is also obvious, and in general I expect ideas in an ML paper to reach a sufficient degree of difficulty for them to be considered a novel contribution for purposes of refereeing.
> >
> > * **Q6.** Thank you, on this particular point this significantly helps, and is exactly the kind of thing I was looking for in terms of soundness.

---

> > > ### Author Response · Authors · 2025-11-21
> > > **Response to Reviewer Azep**
> > >
> > > We appreciate the reviewer's prompt response and feedback. We respond to the remaining concerns about novelty and positioning below.
> > >
> > > >Q1:  I agree with this characterization, but it does not change the gravity of my key concern about novelty and positioning. There are existing papers on offline bandits, mirroring the literature on offline reinforcement learning, but neither the word "bandit" nor the word "reinforcement" is mentioned in the paper - not even offline reinforcement learning in spite of its relevance. Ideas from this literature need to be discussed and benchmarked against, at minimum so that readers know that prior work exists in this area.
> > >
> > > We thank the reviewer for raising this important point.
> > >
> > > At a high level, these two methods can be graphically represented by the following sketch:
> > > ```text
> > > Predict-then-optimize
> > > feature x ------> prediction y ------> decision z
> > >            model            optimization
> > >
> > > Offline contextual bandits (the reduction suggested by the reviewer)
> > > feature x ---------------------------> decision z
> > >                       model
> > > ```
> > >
> > > We agree that these methods are similar in terms of learning a mapping from $X \to Z$, but this high-level view **ignores how DFL benefits from incorporating task-specific optimization and structural information**.
> > >
> > > Specifically, in DFL, the decision $z$ is a solution of solving a known optimization problem, with **explicit constraints and objective** (e.g., quadratic objective with convex constraints). In other words, DFL explicitly incorporates the known structure of the decision task into the learning problem. Predict-then-optimize methods specifically aim to leverage this structure of the optimization problem to learn better predictions under fewer samples. This incorporation of task structure is the key motivation for predict-then-optimize methods.
> > >
> > > In an offline bandit view, each decision (action) $z$ is obtained by a policy that is learned directly without using the fact that $z$ comes from solving a particular optimization problem. In theory, if the model is expressive enough, an offline bandit algorithm can learn the decision with enough samples, but it may ignore the underlying optimization structure and thus require more samples. The pioneering work on DFL (Donti et al., 2017) has shown that DFL is significantly more sample-efficient than a pure policy optimizer that ignores the known optimization problem structure. Also, offline RL baselines consider **sequential decision making**, which is beyond the scope of this paper.
> > >
> > > In summary, we believe the reduction of DFL to the offline contextual bandits setting loses significant structural information about the decision task. However, if the reviewer believes that it would still be useful to implement a direct comparison, we are open to running some experimental comparisons during this discussion period. We are also happy to clarify the distinction between offline contextual bandits and DFL in the paper to help readers with a background in RL/bandits.
> > >
> > > Reference:
> > > - Priya L. Donti, Brandon Amos, J. Zico Kolter. Task-based End-to-end Model Learning in Stochastic Optimization. NIPS, 2017.
> > >
> > > >Q2: What I am asking is for you to re-write your paper to reflect this framing, and present your contributions in an honest and non-misleading way that mirrors your response - namely, that, as you say "the use of diffusion models to formulate the stochastic optimization in the predict-then-optimize framework is new". Your paper is not written in this manner, it needs to be re-written and re-submitted to reflect this.
> > >
> > > We thank the reviewer for your feedback. We think the confusion is coming primarily from our literature review section, where we can do a better job at clearly delineating between the established predict-then-optimize framework and our specific novel contributions. We will revise our draft based on our discussion.

---

> > > > ### Comment · Reviewer_Azep · 2025-11-26
> > > >
> > > > > ignores how DFL benefits from incorporating task-specific optimization and structural information
> > > >
> > > > This is statement is speculation, as in your updated draft there appears to be no theoretical or experimental evidence offered to support it. I would be convinced by an experimental comparison to an offline contextual bandit method, but, frankly, **stating DFL benefits from this in your rebuttal - without appropriate supporting evidence, whether in the form of a theorem or experiment - is misleading**.
> > > >
> > > > > offline RL baselines consider sequential decision making, which is beyond the scope of this paper
> > > >
> > > > I find this choice of scope unconvincing: non-sequential decision making is a special case of sequential decision-making where you consider sequences of length one. In fact, it very natural for readers to wonder what performance would be like relative to a naive baseline like this, precisely because, as you say, it is a more-general method. Your method is specialized so one could expect it to perform better. But it might also be seen as more complex, whereas simpler methods often work best in practice. In situations like this where looking at different aspects leads to different intuitions, what is the right answer? Does it actually perform better in reality on a reasonably-designed baseline? I think it is very important for readers to know.
> > > >
> > > > > We are also happy to clarify the distinction between offline contextual bandits and DFL in the paper to help readers with a background in RL/bandits.
> > > >
> > > > While I respect your acknowledgment of this issue, I checked your revised draft, and this has not been done. Specifically, according to a search, **neither the word "bandit", nor "reinforcement", nor "offline" occur anywhere in your updated text**. This very much reinforces my original judgment that (a) your presentation which omits these related works is misleading, (b) that the changes needed are too large to handle in a rebuttal, and (c) as consequence, it is very important for this paper to be rejected so that they can be incorporated in a resubmitted version. I will therefore maintain my score.

---

> > > > > ### Author Response · Authors · 2025-12-02
> > > > > **Response to Reviewer Azep**
> > > > >
> > > > > Thank you for the suggestion. The only remaining concern is whether a DFL paper should compare with the offline contextual bandit literature. To our knowledge, none of the existing DFL papers discuss offline contextual bandits, precisely because in DFL the objective structure is known, whereas in bandits it is not, so the research questions are fundamentally different.
> > > > >
> > > > > To respectfully address your comment, we've added more evidence and discussion to demonstrate that our proposed diffusion DFL achieves better performance than offline bandit methods. Specifically, we added a policy-based offline contextual bandit baseline in Experiments (Section 6) and also a paragraph in Related Works (Section 2) to clarify the connection between DFL and offline contextual bandits. The changes are marked in blue, and the updated code is here: https://anonymous.4open.science/r/Diffusion_DFL-C2FB/README.md.

---

### Author Response · Authors · 2025-11-20
**Rebuttal Summary**

Here, we summarize our main response to reviewers and highlight key changes in our paper revision:

1. We clarify that our work addresses the **offline** predict-then-optimize setting, which is fundamentally different from **online** contextual bandits. We emphasize that our main contribution is a scalable way to differentiate through diffusion-based stochastic objective.
2. We add Proposition A.7, which provides an upper-bound on the gap between the ELBO gradient and the true log-likelihood gradient in terms of the KL divergence between the variational and true posteriors. If the diffusion model learns an accurate variational posterior distribution, then the gap in gradient approximation is small.
3. We clarify why multi-modal predictive distributions matter in DFL (e.g., electricity demand, long-term financial returns) and add discussion and references showing that two-stage methods (point estimates) are only optimal under restrictive conditions.
4. In response to the reviewers' comments, we also revised our Related Works for a better DFL introduction, added a new discussion of when our Diffusion-DFL will be useful, and made the sample size and other experimental settings more clearly specified.

We thank all the reviewers for their hard work and time in reviewing the paper. Your feedback is extremely helpful for us to improve the paper. Please find our detailed responses to your questions below and we are more than happy to discuss more!

---

### Meta-Review · Area_Chair_pb1R · 2026-01-07

**Summary:**

The reviewers raised the following main concerns. First, reviewer Azep highlighted insufficient discussion and comparison with related work in the offline contextual bandits literature. Second, reviewer dJh7 raised concerns on the computational cost and applicability of the proposed approach, although their overall assessment was positive (with a rating of 8). Additionally, there were concerns on theoretical guarantees and other clarification questions.

**Reviewer Concerns:**

The rebuttal addressed the clarification questions and provided a new theoretical guarantee.

While reviewer dJh7 raised concerns on the computational cost and applicability of the proposed approach, their overall evaluation of the paper was positive (with a score of 8). These concerns are valid, and the corresponding author-reviewer discussion should be reflected more clearly in the revised manuscript. Although the authors indicated that a discussion comparing the proposed method with the two-stage method was added in Section 8, no clearly highlighted modification was observed.

The authors and the reviewer Azep engaged in multiple rounds of discussion. The remaining major concern is whether the paper should discuss and compare against related work in offline contextual bandits. While the studied decision-focused learning (DFL) framework has an additional optimization structure, it shares meaningful similarities with contextual bandit formulations. The AC agrees with Reviewer Azep that offline contextual bandit should be viewed as a general baseline. Following the discussion period freeze due to this year’s incident, the authors added both discussion and empirical comparisons with offline contextual bandits in the revised manuscript.

Overall, the AC recommends weak accept for the revised submission. The AC strongly encourages the authors to incorporate additional and expanded discussions reflecting the feedback from all reviewers, particularly reviewers Azep and dJh7, into the final manuscript.

**Reviewer Scores:**

Had reviewers been able to participate fully in the discussion, the AC believes that Reviewers dJh7, ZKgV, and 1MXz would likely have maintained their original scores. It is less clear whether Reviewer Azep would have revised their score.

---

### Decision · Program_Chairs · 2026-01-26

Accept (Poster)